# Generalization in VAE and Diffusion Models: A Unified Information-Theoretic Analysis

**Qi Chen**[1,3,4,*]**, Jierui Zhu**[2]**, & Florian Shkurti**[1,4,5,*]
[1]Department of Computer Science, University of Toronto
[2]Department of Statistical Sciences, University of Toronto
[3]Data Science Institute, [4]Vector Institute, [5] Robotics Institute

## Abstract

Despite the empirical success of Diffusion Models (DMs) and Variational Autoencoders (VAEs), their generalization performance remains theoretically underexplored, especially lacking a full consideration of the shared encoder-generator structure. Leveraging recent information-theoretic tools, we propose a unified theoretical framework that provides guarantees for the generalization of both the encoder and generator by treating them as randomized mappings. This framework further enables (1) a refined analysis for VAEs, accounting for the generator's generalization, which was previously overlooked; (2) illustrating an explicit trade-off in generalization terms for DMs that depends on the diffusion time $T$; and (3) providing computable bounds for DMs based solely on the training data, allowing the selection of the optimal $T$ and the integration of such bounds into the optimization process to improve model performance. Empirical results on both synthetic and real datasets illustrate the validity of the proposed theory.

## 1 Introduction

Modeling complex data distributions with generative models has become a key focus in machine learning, driving transformative applications in various domains such as computer vision (Rombach et al., 2022), natural language processing (Brown et al., 2020), and scientific discovery (Hoogeboom et al., 2022). Over the past decade, this field has seen rapid advancement with the emerging of frameworks such as Variational Auto-Encoders (VAEs) (Kingma & Welling, 2013; Makhzani et al., 2015; Tolstikhin et al., 2017), Generative Adversarial Networks (GANs) (Goodfellow et al., 2014; Nowozin et al., 2016; Arjovsky et al., 2017), Diffusion Models (DMs) (Song & Ermon, 2019; Song et al., 2020b; 2021), as well as energy-based and auto-regressive models (Larochelle & Murray, 2011; Van den Oord et al., 2016). Notably, deep latent diffusion models that combine VAEs and DMs have recently demonstrated exceptional success in generating high-resolution images (Rombach et al., 2022) and videos (Peebles & Xie, 2023).

The empirical success and appeal of these models lie in their ability to generate new data rather than merely memorizing and reproducing training samples. However, recent studies have demonstrated memorization problems in GANs (Feng et al., 2021; Meehan et al., 2020), VAEs (Van den Burg & Williams, 2021), DMs (Carlini et al., 2023), and auto-regressive Large Language Models (LLMs) (Carlini et al., 2022), raising significant privacy and copyright concerns. Consequently, understanding the generalization ability of generative models in producing diverse, novel samples becomes critical and urgent. While various generalization theories have been developed for GANs (Arora et al., 2017; Biau et al., 2021; Mbacke et al., 2023; Yang & Weinan, 2022; Ji et al., 2021) by adapting different theoretical tools originated in supervised learning (*e.g.*, VC-dimension (Vapnik et al., 1994), Rademacher (Bartlett & Mendelson, 2002), PAC-Bayes(Shawe-Taylor & Williamson, 1997; McAllester, 1998)), their counterparts for VAEs and DMs remain comparatively underexplored.

In this paper, we aim to provide a novel generalization theory for VAEs and DMs. In comparison to previous works, our main contributions are as follows:

---

*Correspondence to: qichen@cs.toronto.edu, florian@cs.toronto.edu.

**1. Unified information-theoretic framework.** VAEs employ a *probabilistic* encoder-generator pair, while DMs can be viewed as an infinite-length composition of such encoder-generator sequences (Tzen & Raginsky, 2019; Huang et al., 2021). We thereby model the encoder and generator as randomized mappings and develop a novel and unified theory with information-theoretic learning tools, avoiding traditional methods designed for deterministic mappings, which are unsuitable for our analysis. The general theoretical results are presented in Sec. 4, where the proposed bounds are algorithm- and data-dependent under the sub-Gaussian assumption.

**2. Improved analysis and tighter bounds for VAEs.** To the best of our knowledge, we are the first to consider the generalization properties of both the encoder and generator in VAEs, whereas Chérief-Abdellatif et al. (2022) only consider guarantees for reconstruction loss and Mbacke et al. (2024) prove bounds for a fixed generator, ignoring its generalization. Moreover, compared to Mbacke et al. (2024), we provide a tighter generalization bound for the encoder by directly bounding the generation error (defined in Sec.3.2), removing the unnecessary Wasserstein-2 distance.

**3. Computable bounds for diffusion models.** We provide non-vacuous upper bounds for DMs to measure the divergence between the generated data and the original data, which can be tractably estimated, as detailed in Theorems 6.2 and 6.3. Through these bounds, we show that:

**(a)** There is an explicit trade-off between the generalization terms of both the encoder and generator that depends on the diffusion time $T$. As presented in Theorem 6.2, when $T$ approaches infinity, the encoder's generalization term vanishes, while the generator's term remains non-zero. Conversely, for small $T$, the encoder's generalization term dominates. Empirical validation on both synthetic and real datasets verifies this phenomenon. Notably, this result implies that *longer diffusion time does not necessarily lead to better generalization.* To the best of our knowledge, this is the first explicit theoretical formulation of this trade-off in the context of generalization theory.

**(b)** The proposed bound provides practical guidance for hyperparameter selection, where previous methods fall short due to the difficulty of accurately estimating the divergences between generated and test data, as shown in Fig. 3. Additionally, the bound can be estimated using only training data, providing a practical and sample-efficient way to select the optimal diffusion time $T$ or integrate the bounds into optimization for better model performance.

## 2 RELATED WORK

In this section, we only discuss the most relevant related works, while other related works on diffusion models and convergence theory are presented in Appendix I.

**Theories for VAEs.** Some previous works (Bozkurt et al., 2019; Huang et al., 2020; Bae et al., 2022) analyze VAEs with rate-distortion theory. Another approach involves deriving exact formulae under specific data distributions and high-dimensional limits. Assuming sample size $m = \infty$, Refinetti & Goldt (2022) examines the test error for nonlinear two-layer autoencoders as $d \to \infty$. Focusing on linear $\beta$-VAEs, Ichikawa & Hukushima (2024) analyzes generalization error with SGD dynamics, where fixed-point analysis reveals posterior collapse when $\beta$ exceeds some threshold, suggesting appropriate KL annealing to accelerate convergence. Ichikawa & Hukushima (2023) uses the Replica method to derive asymptotic generalization error for $\alpha = \frac{m}{d} = \Theta(1)$, showing a peak in error at small $\beta$. This disappears after $\beta$ exceeds some threshold, leading to posterior collapse regardless of the sample size. Husain et al. (2019) build a connection between GAN and WAE, where the generalization is analyzed based on the concentration result of Weed & Bach (2019). Chérief-Abdellatif et al. (2022) applied PAC-Bayes theory to derive the generalization bound for the reconstruction loss. Recently, Mbacke et al. (2024) proved statistical guarantees with PAC-Bayes theory. However, their bounds only consider the generalization properties of the encoder. Instead, we provide tighter bounds for the encoder and use information-theoretic tools to derive generalization bounds for both the encoder and the generator, which are valid for non-linear deep-learning models.

**Generalization theory for diffusion models.** De Bortoli (2022) prove statistical guarantees by simply bounding the Wasserstein-1 distance between the population and empirical data distribution without considering the algorithm or training dynamics. Pidstrigach (2022) discuss the errors in the initial condition and drift terms for SDEs used in the DMs, illustrating the drift explosion under the manifold assumption. They also propose that to avoid purely memorizing data, the exponential integral of the drift approximation error introduced by the score function must be kept infinite when

minimizing the score-matching loss. However, their results are not quantitative. The most recent work (Li et al., 2024) studies the generalization properties of DMs with the random feature model, extending results in Song et al. (2021) by providing separate generalization analysis for the score-matching loss. Its bound cannot capture the trade-off on diffusion time, which was introduced via an ELBO decomposition of the training loss of diffusion models by Franzese et al. (2023). However, our work is the first to show that it exists in generalization, as well as why. Our theoretical results differ from all the mentioned approaches above. We leverage information-theoretic tools to obtain algorithm- and data-dependent bounds under sub-Gaussian assumption, in contrast with the data-dependent bound in Li et al. (2024) that assumes the target data distribution is a 2-mode Gaussian mixture. The proposed bound can provide non-vacuous estimation for the divergence between the original and the generated data distribution, demonstrating an explicit trade-off on the diffusion time.

## 3 PROBLEM SETUP

**Notations.** Detailed notations are summarized in Table 1. We use upper case letters to denote random variables (*e.g.*, $X, Z$) and corresponding calligraphic letters $\mathcal{X}, \mathcal{Z}$ to denote their support sets. We write $\mathscr{P}(\mathcal{X})$ as the set of all the probability measures over $\mathcal{X}$. Then, we denote $P_X \in \mathscr{P}(\mathcal{X})$ as the marginal probability distribution of $X$. Following Husain et al. (2019), we further use $\mathcal{F}(\mathcal{X}, \mathcal{Z}) \stackrel{\text{def}}{=} \{f : \mathcal{X} \to \mathcal{Z}\}$ to denote the set of all the measurable functions from $\mathcal{X}$ to $\mathcal{Z}$. For any $f \in \mathcal{F}(\mathcal{X}, \mathcal{Z})$, the pushforward distribution of $P_X$ through $f$ is denoted as $P_Z^f \stackrel{\text{def}}{=} f \# P_X \in \mathscr{P}(\mathcal{Z})$. Given a Markov chain $X \to Z$, we use $P_{Z|X}$ to represent the distribution over a space $\mathcal{Z}$ conditioned on elements from $\mathcal{X}$, which is also known as the Markov transition kernel from $\mathcal{X}$ to $\mathcal{Z}$. We adopt similar notations for the Markov chain in a reverse direction $Z \to X$ but using $Q_Z, Q_{X|Z}$ to make a distinction. Let $\mathbb{D}(\cdot \| \cdot)$ denote the divergence between two distributions. To be used in our paper, we recall the definitions of Wasserstein distance, the Kullback-Leibler (KL), Jensen-Shannon (JS), and Fisher divergences in Appendix A.2. For any positive integer $m$, we denote $[m] \stackrel{\text{def}}{=} \{1, \ldots, m\}$.

### 3.1 GENERALIZED FORMULATION

Deep generative models typically transform a simple, easy-to-sample prior distribution over a latent space $\mathcal{Z}$ into a target data distribution defined on the input space $\mathcal{X}$ though a generator $G : \mathcal{Z} \to \mathscr{P}(\mathcal{X})$. In most cases, the input and latent spaces are subsets of Euclidean spaces, i.e., $\mathcal{X} \subseteq \mathbb{R}^{d_1}$ and $\mathcal{Z} \subseteq \mathbb{R}^{d_2}$. Ideally, when applied to an easy-to-sample prior distribution $Q_Z$ (*e.g.*, a Gaussian), the optimal generator will induce an identical distribution as the target data distribution $P_X$. However, analogous to the no free lunch theorem in supervised learning (Shalev-Shwartz & Ben-David, 2014), the set of all possible generators $\mathcal{F}(\mathcal{Z}, \mathscr{P}(\mathcal{X}))$ is too complex to be learnable without any prior knowledge, so we further suppose the learning is conducted within a generator hypothesis set $\mathcal{G} \subset \mathcal{F}(\mathcal{Z}, \mathscr{P}(\mathcal{X}))$. The primary goal is to learn a $G$ that matches $Q_X^G \stackrel{\text{def}}{=} G \# Q_Z$ to $P_X$, by minimizing their divergence $\mathbb{D}(P_X \| G \# Q_Z)$. For example, GAN considers the aforementioned goal by directly solving $\inf_{G \in \mathcal{G}} \mathbb{D}_{JS}(P_X \| G \# Q_Z)$.

However, directly learning $G$ may be hard and instable for some data distributions. In this paper, we focus on VAEs and diffusion models, which indirectly learn $G$ as the inverse process of an additional probabilistic encoder $E$, sharing a similar encoder-generator paradigm.

**Encoder-Generator Structure.** Let us define the encoder hypothesis set $\mathcal{E} \subset \mathcal{F}(\mathcal{X}, \mathscr{P}(\mathcal{Z}))$. Then, the encoder is a function $E : \mathcal{X} \to \mathscr{P}(\mathcal{Z}), E \in \mathcal{E}$ that maps an input data point $X \sim P_X$ to a conditional distribution over the latent space $\mathcal{Z}$, *i.e.*, $E(X)$ can be alternately denoted as $P_{Z|X}^E$ at the population level. We further denote the probability densities of $E(X)$ and $P_Z^E \stackrel{\text{def}}{=} E \# P_X$ as $p_E(z|x)$ and $p_E(z)$, respectively. Similarly, the respective probabilistic densities of $G(Z) = Q_{X|Z}^G$ and $G \# Q_Z$ are denoted as $q_G(x|z)$ and $q_G(x)$. The above definitions cover the original auto-encoder, where $E$ is a deterministic encoder by restricting $\mathcal{E}$ as the set of delta distributions. Since $\mathbb{D}_{KL}(P_X \| G \# Q_Z) \leq \inf_{E \in \mathcal{E}}[\mathbb{D}_{KL}(P_X \times E(X) \| Q_Z \times G(Z))]$ with data processing inequality, the objective can be relaxed to solve the upper bound:

$$\inf_{G \in \mathcal{G}, E \in \mathcal{E}} \mathbb{D}_{KL}(P_X \times E(X) \| Q_Z \times G(Z)). \tag{1}$$

Furthermore, we show in Appendix B that VAEs and score-based DMs, respectively, optimize this objective's two forms of decomposition.

### 3.1.1 VARIATIONAL AUTO-ENCODER (VAE)

Based on the general objective, VAE uses a decomposition that is equivalent to the common variational inference approach (Kingma et al., 2019), where the optimization objective is proportional to:

$$\inf_{G \in \mathcal{G}, E \in \mathcal{E}} \left[ \mathbb{E}_{X \sim P_X} \left( \mathbb{E}_{Z \sim E(X)} \left[ -\log q_G(X|Z) \right] + \mathbb{D}_{KL}(E(X) \| Q_Z) \right) \right] . \tag{2}$$

Constraining $E$ and $G$ to specific distribution families leads to the traditional VAE objective. In VAEs, the latent space typically has a much lower dimensionality than the input space, with $d_2 \ll d_1$.

### 3.1.2 DIFFUSION MODEL (DM)

As discussed in (Huang et al., 2021; Tzen & Raginsky, 2019; Kingma et al., 2021), one can treat DMs as infinitely deep hierarchical VAEs by sequentially composing $N$ probabilistic encoders $E_{1:N} \stackrel{\text{def}}{=} \{E_k\}_{k=1}^{N}$ and generators $G_{1:N} \stackrel{\text{def}}{=} \{G_k\}_{k=1}^{N}$ where $N \to \infty$. In another view, we can consider the encoders and generators as time-dependent randomized mappings that directly applied to the original data distribution and the latent prior distribution, respectively. The difference is shown in Fig. 1. We further provide a detailed comparison of these two viewpoints in Appendix B.2 for hierarchical VAEs and in Appendix B.3 for the other.

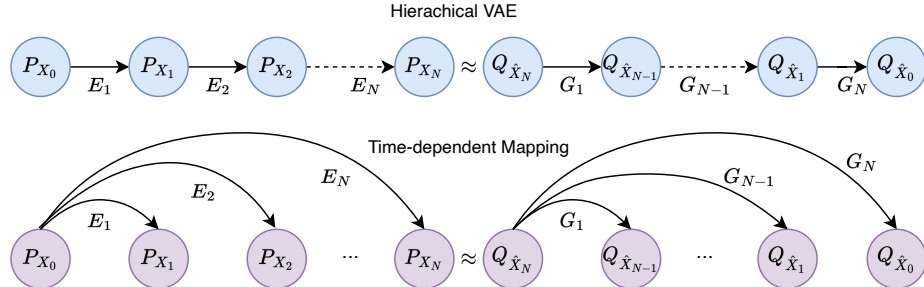

Figure 1: Illustration of DMs as hierarchical VAEs and time-dependent randomized mapping

**Discrete-time stochastic process.** Without loss of generality, we consider the latent space the same as the input space with $d_1 = d_2 = d$ for typical DMs, where $E_k : \mathcal{X} \to \mathscr{P}(\mathcal{X}), G_k : \mathcal{X} \to \mathscr{P}(\mathcal{X})$. For a forward *discrete-time stochastic process*, we denote the marginal distribution at step $k$ as $P_{X_k}$. Then, we have $\forall k \in [N], X_k \sim E_k(X_0), X_0 \sim P_X$, and $P_{X_k} = E_k \# P_{X_0}$, where $P_{X_0} = P_X$ is the initial data distribution. The forward process is often set to achieve some easy-to-sample noise distribution $\pi$ (*e.g.*, $\pi = \mathcal{N}(\mathbf{0}, \mathbf{I}_d)$), where $P_{X_N} \approx \pi$, and $P_{X_N} \xrightarrow{N \to \infty} \pi$ almost surely. Conversely, the backward process starts from $Q_{\hat{X}_N} = \pi$ and aims to achieve $Q_{\hat{X}_0} \approx P_X$. Then, we denote the marginal distribution at step $k$ for the backward process introduced by the the generator sequence $G_{1:N}$ as $Q_{\hat{X}_{N-k}} = G_k \# Q_{\hat{X}_N}$.

**Continuous-time diffusion process.** As in Song et al. (2020b), in the continuous-time limit, the above forward process can form a *diffusion process* $\{X_t\}_{t=0}^{T}$ solving the following SDE over diffusion time T:

$$dX_t = f(X_t, t)dt + \lambda(t)dW_t, X_0 \sim P_X , \tag{3}$$

where we have $P_{X_t} = E_t \# P_{X_0}$, $f(\cdot, t) : \mathcal{X} \to \mathcal{X}$ is the drift coefficient, $\lambda(t) \in \mathbb{R}$ is the diffusion coefficient, and $\{W_t\}_{t \in [0,T]}$ is a Wiener process. With the appropriate selections of $f$ and $\lambda$, the above SDE can converge to the predefined prior distribution $\pi$.

According to Anderson (1982); Haussmann & Pardoux (1986), there exists a reverse-time diffusion process $\{\overleftarrow{X}_t\}_{t \in [0,T]}$ satisfying $\{X_t\}_{t=0}^{T} = \{\overleftarrow{X}_t\}_{t=0}^{T}$ under mild conditions, which is the solution from $t = T$ to $t = 0$ of the following SDE:

$$d\overleftarrow{X}_t = [f(\overleftarrow{X}_t, t) - \lambda(t)^2 \nabla \log p_t(\overleftarrow{X}_t)]dt + \lambda(t)d\overleftarrow{W}_t, \overleftarrow{X}_T \sim P_{X_T} . \tag{4}$$

The above **ideal backward process** can be used to generate reference sample sequences to learn the generative dynamics, and we denote the ideal generator as $E_t^{-1}, \forall t \in [0, T]$. Then, by approximating $\nabla \log p_t(\overleftarrow{X}_t)$ with $\nabla \log q_t(\hat{X}_t)$, the generator $G_t, \forall t \in [0, T]$ is characterized by the following SDE:

$$d\hat{X}_t = [f(\hat{X}_t, t) - \lambda(t)^2 \nabla \log q_t(\hat{X}_t)]dt + \lambda(t)d\hat{W}_t, \hat{X}_T \sim \pi. \tag{5}$$

Define $Q_{\hat{X}_{T-t}} \stackrel{\text{def}}{=} G_t \# \pi$ as the generated distribution at time $t$. Song et al. (2020b; 2021) further proved that, under some regularity conditions, the KL-divergence between the real data distribution $P_X$ and the final-time generated distribution $G_T \# \pi$ is bounded by:

$$\mathbb{D}_{KL}(P_X \| G_T \# \pi) \leq \frac{1}{2} \int_{t=0}^{T} \lambda^2(t) \mathbb{D}_{Fisher}(P_{X_t} \| Q_{\hat{X}_t}) dt + \mathbb{D}_{KL}(P_{X_T} \| \pi),$$

where the Fisher divergence is defined in Def. A.2. As we show in Appendix B, this is an upper bound of another possible decomposition of the general objective in Eq. (1).

## 3.2 Setup for Generalization Analysis

So far, we have discussed the learning objectives of VAEs and DMs at the population level. However, due to the unknown nature of $P_X$, these learning objectives can only be estimated with a training dataset $S = \{X_i\}_{i=1}^m$ of $m$ examples, where each $X_i \sim P_X$ and $S \sim P_X^m$. The empirical distribution of these $m$ observations is then denoted as $\hat{P}_X \stackrel{\text{def}}{=} \frac{1}{m} \sum_{i=1}^m \delta_{X_i}$. By optimizing the encoder $E$ and generator $G$ w.r.t empirical learning objectives, they are learned from the data and mutually dependent. Intuitively, One can as consider them as respectively approximating the posteriors $P_{Z|X,S}$ and $Q_{\hat{X}|Z,S}$.

The encoder-generator process may overfit the empirical distribution $\hat{P}_X$, particularly when the number of training examples $m$ is limited. To measure the performance gap between the population and empirical objectives for generative models, we further define the loss for the encoder-generator pair as $\Delta_G : \mathcal{X} \times \mathcal{Z} \times \mathcal{X} \to \mathbb{R}_0^+$. In particular, we use $\Delta_G(\hat{X}, Z, X) = \|\hat{X} - X\|$ in Corollary. 4.2 and $\Delta_G(\hat{X}, Z, X) = -\log q_G(X|Z)$ in Corollary. 4.3.

Let us first consider the **empirical reconstruction error** [1], by which we can measure the input-output distortion in expectation to the empirical measure $\hat{P}_X$ of the train dataset $S$:

$$\mathcal{L}_{\hat{P}_X}(E, G) \stackrel{\text{def}}{=} \frac{1}{m} \sum_{i=1}^m \mathbb{E}_{Z \sim E(X_i)} \mathbb{E}_{\hat{X} \sim G(Z)} \Delta_G(\hat{X}, Z, X_i).$$

Now, we define the **generation error** as the expected difference between any input $X$ sampled from the data distribution $P_X$ and any output generated by $G(Z)$ with $Z$ being sampled from the prior $\pi$:

$$\mathcal{L}_{P_X}^\pi(E, G) \stackrel{\text{def}}{=} \mathbb{E}_{X \sim P_X} \mathbb{E}_{Z \sim \pi} \mathbb{E}_{\hat{X} \sim G(Z)} [\Delta_G(\hat{X}, Z, X)].$$

The dependence on encoder $E$ is implicit and specific to the encoder-generator paradigm, where the learning of $G$ relies on $E$. Based on the definitions above, we introduce the **generalization gap** that measures the difference between the generation error and the empirical reconstruction error:

$$gen_{P_X}^{\Delta_G}(E, G, \pi) \stackrel{\text{def}}{=} \mathbb{E}_{S \sim P_X^m} \left[ \mathcal{L}_{P_X}^\pi(E, G) - \mathcal{L}_{\hat{P}_X}(E, G) \right].$$

Intuitively, when the encoder and generator are learned from an empirical reconstruction process with very small reconstruction error (*i.e.*, $\mathcal{L}_{\hat{P}_X}(E, G)$ is small), the generalization gap reflects the average difference between the generated and original data.

## 4 General Theoretical Results

In this section, we present general theoretical results for generative models that share the same encoder-generator paradigm, which can be directly extended to analyze models like VAEs and DMs.

---
[1]Similar notion as distortion can be found in Blau & Michaeli (2019).

**Theorem 4.1.** *For any encoder $E \in \mathcal{E}$ and generator $G \in \mathcal{G}$ learned from the training data $S = \{X_i\}_{i=1}^{m}$, assume that the loss $\Delta_G(\tilde{\hat{X}}, \tilde{Z}, X)$ is R-sub-Gaussian (See definition in Def. A.4) under $P_{\tilde{\hat{X}}, \tilde{Z}, X} = Q_{\hat{X}|Z} \times Q_Z \times P_X$, where $Z \sim Q_Z = \pi, \hat{X} \sim G(Z), \tilde{\hat{X}}, \tilde{Z}$ are respective independent copy of $\hat{X}$ and $Z$ such that $\tilde{\hat{X}}, \tilde{Z} \perp\!\!\!\perp X$. Then, $\forall X_i \in S, Z_i \sim E(X_i), \hat{X}_i \sim G(Z_i)$, the generalization gap admits the following bound:*

$$|gen_{P_X}^{\Delta_G}(E, G, \pi)| \leq \frac{\sqrt{2}R}{m} \sum_{i=1}^{m} \sqrt{\mathbb{E}_{X_i}[\mathbb{D}_{KL}(E(X_i)\|\pi)] + I(\hat{X}_i; X_i|Z_i)}.$$

**Discussion.** Since $\sqrt{a+b} \leq \sqrt{a} + \sqrt{b}, \forall a, b > 0$, the above bound can be further decomposed as $\frac{\sqrt{2}R}{m} \sum_{i=1}^{m} \sqrt{\mathbb{E}_{X_i}[\mathbb{D}_{KL}(E(X_i)\|\pi)]} + \frac{\sqrt{2}R}{m} \sum_{i=1}^{m} \sqrt{I(\hat{X}_i; X_i|Z_i)}$. The first term measures, on expectation of randomly drawing $m$ data points, the average divergence from their encoded latent distributions to the predefined prior $\pi$, which reflects the generalization of the encoder. The condition mutual information in the second term $I(\hat{X}_i; X_i|Z_i)$ measures the generalization of the generator $G$.

This bound provides insight into how the generalization of the encoder and generator interact. Assuming the reconstruction error is small: If the first term approaches zero, *i.e.*, $E(X_i) = \pi$, which means $Z_i$ contains no information of the training data, the generalization gap is entirely attributed to the generator. In contrast, if the encoder overfits to the training data and $Z_i$ fully captures the information from $X_i$, then $I(\hat{X}_i; X_i|Z_i) = 0$, making the second term zero. In this case, the generalization gap is entirely due to the encoder. In intermediate scenarios, both the encoder and generator contribute to the overall generalization. The detailed proof can be found in Appendix C.1.

By specifying $\Delta_G$, we have the following two corollaries that measure the divergence between the true data distribution $P_X$ and the generated distribution $G\#\pi$, as formulated in Sec. 3.1.

**Corollary 4.2.** *Under Theorem 4.1, let $\Delta_G(\hat{X}, Z, X) = \|\hat{X} - X\|$. Then, the Wasserstein distance between the data distribution $P_X$ and the generated distribution $G\#\pi$ is upper bounded by:*

$$\mathbb{D}_{W_1}(P_X\|G\#\pi) \leq \mathbb{E}_S \mathcal{L}_{\hat{P}_X}(E, G) + \frac{\sqrt{2}R}{m} \sum_{i=1}^{m} \sqrt{\mathbb{E}_{X_i}[\mathbb{D}_{KL}(E(X_i)\|\pi)] + I(\hat{X}_i; X_i|Z_i)}.$$

**Corollary 4.3.** *Under Theorem 4.1, let the density function of probabilistic decoder $G$ given a latent code $z$ be $q_G(\cdot|z) : \mathcal{X} \to \mathbb{R}_0^+$ and $\Delta_G(\hat{X}, Z, X) = -\log q_G(X|Z)$. The KL-divergence between the data distribution $P_X$ and the generated distribution $G\#\pi$ is then upper bounded by:*

$$\mathbb{D}_{KL}(P_X\|G\#\pi) \leq \mathbb{E}_S \mathcal{L}_{\hat{P}_X}(E, G) + \frac{\sqrt{2}R}{m} \sum_{i=1}^{m} \sqrt{\mathbb{E}_{X_i}[\mathbb{D}_{KL}(E(X_i)\|\pi)] + I(\hat{X}_i; X_i|Z_i)} - h(P_X),$$

*where $h(P_X) = \mathbb{E}_X[-\log p(X)]$ denotes the entropy.*

The proofs of these two corollaries are presented in Appendix C.2. In following sections, we will apply Corollary 4.2 and Corollary 4.3 to establish Theorem 5.1 and Theorem 6.2, respectively.

## 5 ANALYSIS OF VAEs

VAEs specify tractable distributions to both encoder and generator. Normally, they are set as Gaussians, as presented in the original VAE (Kingma & Welling, 2013). Using some neural networks parametrized by $\phi$, one can map the original data to the latent space and model both the mean and variance of the Gaussian as $\mu_\phi : \mathcal{X} \to \mathcal{Z}$ and $\sigma_\phi : \mathcal{X} \to \mathcal{Z}$. The encoder is then $E_\phi(x) = \mathcal{N}(\mu_\phi(x), \text{diag}(\sigma_\phi^2(x))\mathbf{I}_{d_2})$. Analogously, the generator network is parameterized by $\theta$, which often only models the mean, where $G_\theta(z) = \mathcal{N}(\mu_\theta(z), \mathbf{I}_{d_1})$ with $\mu_\theta : \mathcal{Z} \to \mathcal{X}$. Directly applying Corollary 4.2, we can obtain the following bound for VAE:

**Theorem 5.1.** *Under the assumptions made in Theorem 4.1 and Corollary 4.2, we have $\forall X_i \in S$, $Z_i \sim E_\phi(X_i), \hat{X}_i \sim G_\theta(Z_i)$ over the draw of $m$ samples with $S \sim P_X^m$ that the following bound holds for any probabilistic encoder $E_\phi$ and generator $G_\theta$ defined above:*

$$\mathbb{D}_{W_1}(P_X\|G_\theta\#\pi) \leq \mathbb{E}_S \mathcal{L}_{\hat{P}_X}(E_\phi, G_\theta) + \frac{\sqrt{2}R}{m} \sum_{i=1}^{m} \sqrt{\mathbb{E}_{X_i}[\mathbb{D}_{KL}(E_\phi(X_i)\|\pi)] + I(\hat{X}_i; X_i|Z_i)}.$$

**Comparison with previous VAE bounds.** The above bound could be compared to the recent PAC-Bayes bound for VAE either by converting to a high-probability bound using the results of Theorem 3 in Xu & Raginsky (2017) or by converting PAC Bayes bound to its expectation version. In sharp contrast to the bound in Theorem 5.2 of Mbacke et al. (2024), which applies only to a fixed $\theta$, the above result guarantees generalization for any generator $G_\theta$ with $\frac{1}{m}\sum_{i=1}^{m} I(\hat{X}_i; X_i|Z_i)$. Additionally, our approach avoids introducing an extra Wasserstein-2 distance, as we directly bound the generation error using the empirical reconstruction loss. In contrast, Mbacke et al. (2024) utilize the triangle inequality for the Wasserstein distance, separately bounding $\mathbb{D}_{W_1}(P_X \| G_\theta \# \hat{P}_Z^{E_\phi})$ and $\mathbb{D}_{W_1}(G_\theta \# \hat{P}_Z^{E_\phi} \| G_\theta \# \pi)$. Furthermore, we relax the strict assumption of bounded support, replacing it with the more flexible sub-Gaussianity condition. Detailed mathematical and experimental comparisons are respectively provided in Sec. D.1 and Sec G.1 in the Appendix, showing the improvements.

**Insights and practical guidance for VAEs.** If we instead apply Corollary 4.3, the reconstruction error becomes $\mathcal{L}_{\hat{P}_X}(E_\phi, G_\theta) = \frac{1}{m}\sum_{i=1}^{m} \mathbb{E}_{Z_i \sim E_\phi(X_i)}[-\log q_{G_\theta}(X_i|Z_i)]$. Interestingly, jointly optimizing it with $\frac{1}{m}\sum_{i=1}^{m} D_{KL}(E(X_i)\|\pi)$ yields the empirical estimate of the VAE objective in Eq. (2). This implies that the VAE training process inherently accounts for the encoder's generalization. However, the generalization of the generator $G$ is often overlooked. Therefore, a potential improvement could involve explicitly incorporating the generator's generalization into the optimization objective as a regularization term. In Appendix D.2, we derive an example regularizer using upper bound of the conditional mutual information term by introducing an additional randomly initialized generator.

## 6 ANALYSIS OF DIFFUSION MODELS

In existing literature, most works focus on the convergence of DMs, while the limited analysis of their generalization properties typically involves separately bounding $\mathbb{D}(P_X\|\hat{P}_X)$ and $\mathbb{D}(\hat{P}_X\|G_T\#\pi)$ using concentration results, then combining them via the triangle inequality. However, widely used divergences in diffusion models, such as KL-divergence, do not satisfy the triangle inequality. Moreover, these bounds are often loose due to strong assumptions about the data distribution, score function estimation, and insufficient consideration of the learning algorithm. To address these, we directly bound $\mathbb{D}(P_X\|G_T\#\pi)$ and apply information-theoretic tools to derive computable algorithm- and data-dependent bounds in this section.

### 6.1 GENERATION ERROR BOUND FOR SCORE-BASED DMs

For DMs, the encoders and generators $E_t, G_t, \forall t \in [0, T]$ are restricted to the family of SDEs or the discretized Langevin Dynamics. Before bounding the generation error, we first prove the following lemma (proof in Appendix E.1.) on the empirical reconstruction error at the diffusion end time $T$:

**Lemma 6.1.** *Let $\{X_t\}_{t=0}^{T}$ be the empirical version of the forward diffusion process defined in Eq. (3), where $X_0 \sim \hat{P}_X$. We assume the existence of the backward process under the regularity conditions outlined in Song et al. (2021) and denote it as $\{\overleftarrow{X}_t\}_{t=0}^{T} = \{X_t\}_{t=0}^{T}$, which results from the reverse-time SDE defined in Eq. (4). Then, the generative backward process $\{\hat{X}_t\}_{t=0}^{T}$ is defined in Eq. (5). Let $E_t, E_t^{-1}, G_t, \forall t \in [0, T]$ be their corresponding time-dependent Markov kernels. The density function of any generator $G$, given a latent code $z$, is denoted as $q_G(\cdot|z) : \mathcal{X} \to \mathbb{R}_0^+$, and let $\Delta_G(\hat{X}, Z, X) = -\log q_G(X|Z)$. Then, we have*

$$|\mathcal{L}_{\hat{P}_X}(E_T, G_T) - \mathcal{L}_{\hat{P}_X}(E_T, E_T^{-1})| \leq \frac{1}{2}\int_{t=0}^{T} \lambda^2(t) \mathbb{D}_{Fisher}(\hat{P}_{X_t}\|Q_{\hat{X}_t})dt.$$

**Connection to score matching.** By setting the derivative of the log density of $Q_{\hat{X}_t}$ with some parameterized function, *i.e.*, $\nabla_x \log q_t(x) = s_\theta(x, t)$, the upper bound in the above theorem gives the following empirical loss of Explicit Score Matching (ESM):

$$\hat{\mathcal{L}}_{ESM}(\theta, \lambda(\cdot)) = \frac{1}{2}\int_{t=0}^{T} \mathbb{E}_{X_t \sim \hat{P}_{X_t}}[\lambda^2(t)\|\nabla_{X_t}\log\hat{p}_t(X_t) - s_\theta(X_t, t)\|_2^2]dt.$$

Since $\hat{p}_t(X_t) = \frac{1}{m}\sum_{i=1}^{m} p_{E_t}(X_t|X_i)$, it gives the following Denoising Score Matching (DSM) loss:

$$\hat{\mathcal{L}}_{DSM}(\theta, \lambda(\cdot)) = \frac{1}{2}\int_{t=0}^{T} \mathbb{E}_{X_0 \sim \hat{P}_X, X_t \sim E_t(X_0)}[\lambda^2(t)\|\nabla_{X_t}\log p_{E_t}(X_t|X_0) - s_\theta(X_t, t)\|_2^2]dt,$$

which is equivalent to ESM up to some constant as discussed in (Vincent, 2011). Combining Corollary 4.3 and Lemma 6.1, we have the following generation error bound for score-based diffusion models:

**Theorem 6.2.** *Under Lemma 6.1, for any SDE encoder $E_t$ and generator $G_t^\theta$ trained via score matching on $S = \{X_i\}_{i=1}^m$, the corresponding outputs at the diffusion time $T$ are $\hat{X}_T \sim E_T(X_i)$, $\hat{X}_0 \sim G_T^\theta(\hat{X}_T)$ for each $X_i \in S$. The KL-divergence between the original data distribution $P_X$ and the generated data distribution $G_T^\theta \# \pi$ at diffusion time $T$ is then upper bounded by:*

$$\mathbb{D}_{KL}(P_X \| G_T^\theta \# \pi) \le \mathbb{E}_S \underbrace{\left( -\frac{1}{m} \sum_{i=1}^m \mathbb{D}_{KL}(E_T(X_i) \| E_T \# \hat{P}_X) + \hat{\mathcal{L}}_{ESM}(\theta, \lambda(\cdot)) \right)}_{T_1}$$

$$+ \underbrace{\frac{\sqrt{2}R}{m} \sum_{i=1}^m \sqrt{\mathbb{E}_{X_i}[\mathbb{D}_{KL}(E_T(X_i) \| \pi)]}}_{T_2} + \underbrace{\frac{\sqrt{2}R}{m} \sum_{i=1}^m \sqrt{I(\hat{X}_0; X_i | \hat{X}_T)}}_{T_3}.$$

The proof is presented in Appendix E.2. Compared to recent work on the generalization of DMs (Li et al., 2024), we demonstrate the existence of a trade-off w.r.t the diffusion time $T$. In contrast to Franzese et al. (2023) that also mentioned the diffusion time trade-off, we prove an explicit form of the tradeoff related to generalization terms, whereas Franzese et al. (2023) only justified it via a new ELBO decomposition of the training loss at the population level, without being able to show how it affects generalization.

**Explicit trade-off on diffusion time $T$.** The KL-divergence terms in the above bound reflect the generalization of encoder $E_T$. Since $T_1 < 0$ and $T_2 > 0$, there exists an inherent trade-off on the diffusion time $T$ when minimizing the two terms. Note that when $T \to \infty$, the forward SDE maps the empirical data distribution to the noise, which means $E_T \# \hat{P}_X$ will converge to $\pi$. This makes the two KL terms $T_1$ and $T_2$ in the above theorem equivalent, and both will converge to zero, as discussed in Sec 6.2. However, $T_3$, which characterizes the generalization of generator $G_T$, will remain non-zero for a small sample size $m$. We bound $T_3$ in Sec. 6.3 to a easy-to-compute form, showing a linear growth w.r.t $T$. This means another trade-off between the generalization of the encoder and that of the decoder exists on diffusion time.

## 6.2 GENERALIZATION FOR SDE ENCODERS

The typical encoder of diffusion models is set to a special class of affine SDEs that has a closed-form solution (Särkkä & Solin, 2019), where $dX_t = \alpha(t)X_t dt + \lambda(t)dW_t$. Then, the encoder posterior for a given example equals to $E_T(X_i) = \mathcal{N}(r(T)X_i, r^2(T)v^2(T)\mathbf{I}_d)$, where $r(t) = e^{\int_0^t \alpha(t')dt'}$, $v(t) = \sqrt{\int_0^t \lambda^2(t')/r^2(t')dt'}$.

**Variance-exploding SDEs** with parameter $\alpha(t) = 0$, $\lambda(t) = \sqrt{d\sigma^2(t)/dt}$, $\sigma^2(t) = (\sigma_{max}^2/\sigma_{min}^2)^t$ have $E_T(X_i) = \mathcal{N}(X_i, (\sigma^2(T) - \sigma^2(0))\mathbf{I}_d)$, which do not converge to a steady-state distribution because the variance grows. Setting the prior as $\pi = \mathcal{N}(\mathbf{0}, (\sigma^2(T) - \sigma^2(0))\mathbf{I}_d)$, we have

$$\mathbb{D}_{KL}(E_T(X_i) \| \pi) = \tfrac{1}{2} \left( X_i^T X_i / (\sigma^2(T) - \sigma^2(0)) \right) \xrightarrow{T \to \infty} 0.$$

**Variance-preserving SDEs** converge to multivariate Gaussians with $\alpha(t) = -\frac{1}{2}\lambda^2(t)$, $\lambda(t) = \sqrt{\beta_0 + (\beta_1 - \beta_0)t}$. By denoting $\beta_T = \beta_0 T + \frac{1}{2}(\beta_1 - \beta_0)T^2$, the encoder posterior equals to $E_T(X_i) = \mathcal{N}(e^{-\frac{1}{2}\beta_T}X_i, (1 - e^{-\beta_T})\mathbf{I}_d)$. Setting the prior as $\pi = \mathcal{N}(\mathbf{0}, \mathbf{I}_d)$, we have

$$\mathbb{D}_{KL}(E_T(X_i) \| \pi) = \tfrac{1}{2} \left( e^{-\beta_T} X_i^T X_i - de^{-\beta_T} - d\log(1 - e^{-\beta_T}) \right) \xrightarrow{T \to \infty} 0.$$

## 6.3 GENERALIZATION FOR DISCRETIZED SDE GENERATORS

We focus on the generalization terms and drop the analysis of the convergence w.r.t the score-matching training process, which has been done in Li et al. (2024). The generalization gap term related to the generator $G_T^\theta$ is determined by $I(\hat{X}_0, X_i | \hat{X}_T)$, where $\theta$ is learned from data and can be represented as some function of the train dataset.

**Theorem 6.3.** *Let the step size be $\tau = \frac{T}{N}$, where we split $T$ to $N$ discrete times. For any $k \in [N]$, we use the following discrete update for the backward SDE by setting $\epsilon_{t_k} \sim \mathcal{N}(0, \mathbf{I}_d), t_k = T - \tau k$:*
$$\hat{X}_{t_k} = (1 - \frac{\tau}{2}\lambda^2(T - t_{k-1}))\hat{X}_{t_{k-1}} + \tau\lambda^2(T - t_{k-1})s_\theta(X_{t_{k-1}}, T - t_{k-1}) + \sqrt{\tau}\lambda(T - t_{k-1})\epsilon_{t_{k-1}}.$$
*Furthermore, we assume a bounded score $\|\nabla_x \log \hat{p}_t(x)\| \leq L, \forall x, t$. Then, we have*

$$\frac{1}{m}\sum_{i=1}^{m} I(\hat{X}_0; X_i | \hat{X}_T) \leq \frac{1}{m} I(\hat{X}_0; X_{1:m} | \hat{X}_T) \leq \frac{TL^2 \sum_{k=1}^{N} \lambda^2(\frac{(k-1)T}{N})}{2mN}.$$

The proof is deferred to Appendix F. This theorem can be used to estimate $T_3$, which has a linear growth w.r.t $T$ for variance preserving SDE. Combining Theorem 6.2, we obtain the sample complexity $\mathcal{O}(1/\sqrt{m})$, compared to $\mathcal{O}(m^{-2/5})$ in Li et al. (2024) using the random feature model.

**Practical guidance for DMs.** Since the upper bound in Theorem 6.2 can be estimated with only training data, we can train the model for various diffusion times, estimate the bound, and **select the optimal** $T$ via grid search. Additionally, the generalization terms in the theorem can be incorporated as regularization when optimizing the score-matching model parameter $\theta$. This can be achieved by selecting appropriate values for $\beta_0, \beta_1$ in $\lambda(t)$, or by adding a gradient penalty to control the Lipschitz constant of the score model $s_\theta$.

## 7 EXPERIMENTS

In this section, we focus on validating Theorem 6.2 for score-based DMs using both synthetic and real datasets. The numerical results illustrate the existence of the trade-off between the generalization terms of encoder and generator on diffusion time, which significantly impacts generation performance. Experiments for Theorem 5.1 of VAEs are deferred to Sec G.1 in the Appendix, where we show the proposed bound can better capture the generalization of VAEs compared to previous bounds with the additional mutual information term for generator, and its relation to the memorization score proposed in Van den Burg & Williams (2021).

### 7.1 SYNTHETIC DATA

We begin by validating the theorem on a simple synthetic 2D dataset derived from the Swiss Roll dataset. We train the score matching model $s_\theta(x, t)$ and estimate the upper bound in Theorem 6.2 on a training set of size $m$. W.r.t the expectation over dataset $S$, we conduct 5-times Monte-Carlo estimation by randomly generating train datasets with different random seeds. For the left-hand-side KL-divergence, we conduct Monte Carlo estimation of with 1000 test data points. To get more details on the estimation, please see Appendix G.2.1.

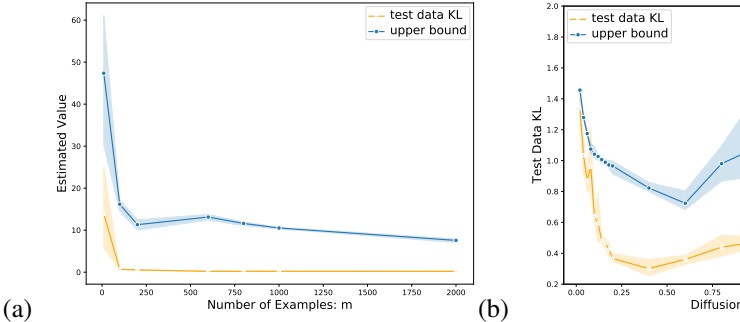
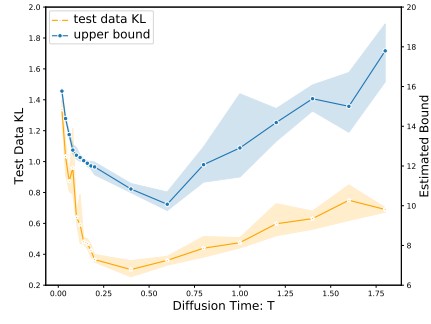

(a)  (b)

Figure 2: Evolution of bound and test-data KL-divergence estimation w.r.t (a) train dataset size $m$ when diffusion time is $T = 1$, (b) diffusion time $T$ when the training dataset size is $m = 200$.

**Sample complexity.** We can observe in Fig. 2(a) that both the estimated test-data KL divergence and the upper bound decrease with the increase in train dataset size $m$, corresponding to the diminish of $T_3$ with order $\mathcal{O}(1/\sqrt{m})$ as $m \to \infty$. However, they will not converge to zero, which is due to the diffusion time $T = 1 \neq \infty$ with non-zero $T_1, T_2$ and the optimization error of score matching loss. The quality of generated data for models trained on different sample size $m$ aligns with human perception, as presented in Fig. 5 in Appendix.

**Trade-off on diffusion time.** In Fig. 2 (b), we can observe the trade-off on diffusion time $T$ on both the bound and the estimated KL divergence. This indicates the proposed bounds are non-vacuous and can capture the algorithm and data distribution well. In addition, the optimal diffusion time lies in the range of 0.4 to 0.6. We visualize the generated data for each diffusion time in Fig. 6 in the Appendix, and we can observe the generated data points that fit the test data best in human perception also take values at $T = 0.4$ or $T = 0.6$.

## 7.2 REAL DATA

We further estimate the bound and the test data KL divergence (or log densities) by training DMs on MNIST and CIFAR10 datasets with few-shot data ($m = 16$) and full train dataset. For the full data setting, in Fig. 3 (c), we can observe the trade-off on both the estimated bound using training data and log-likelihood (in bit per dimension (BPD)) estimated on 10000 test data points. For the few-shot scenario, we observe a trade-off between noise and duplicate (or entirely black/white) images in the generated data with respect to the diffusion time $T$ across both datasets, as shown in Fig. 3 (a). In Fig.3 (b), the estimated bound can verify the diffusion time trade-off, where the optimal $T$ is around 0.8 for both datasets, more detailed results are presented in Appendix (Fig. 9 and Fig. 10). However, the test data KL divergence and log-likelihood do not reflect the trade-off. This is consistent with the conclusion in Theis et al. (2015) that it's challenging to obtain accurate estimation of the KL divergence and the BPD for high-dimensional data distribution with limited data.

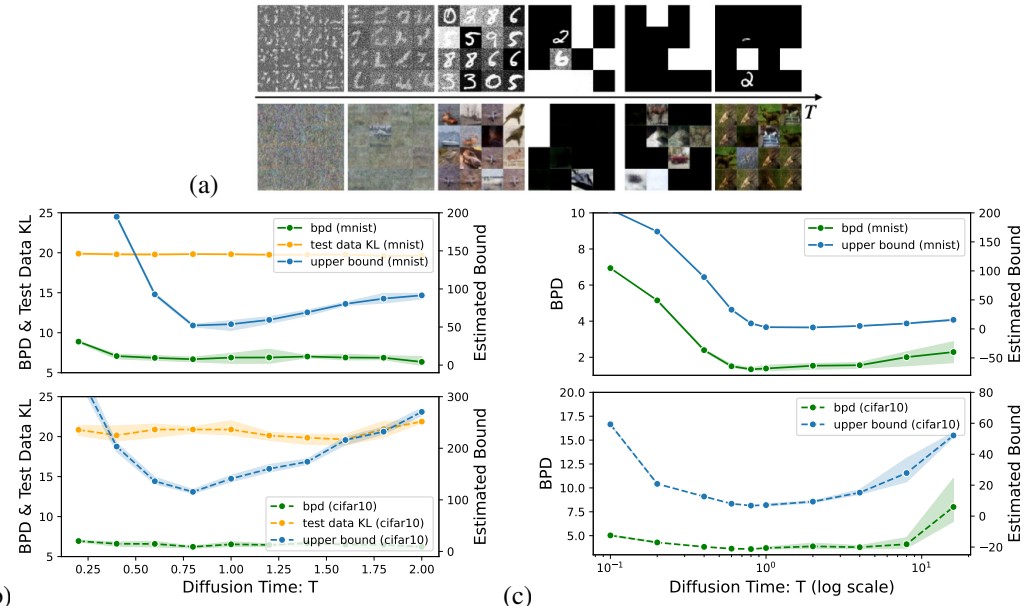

Figure 3: (a) For DM trained on few-shot MNIST and CIFAR10 data ($m = 16$): The trade-off on diffusion time reflects on the generated images with the growing of $T$; (b) The bounds estimated on train data, KL divergences, and log densities (bpd–bits per dimension) estimated on 100 test samples. (c) The bound estimated on train data and log densities estimated on 10000 test samples for DM trained on full MNIST ($m = 60000$) and CIFAR10 dataset ($m = 50000$).

## 8 CONCLUSION

In this work, we provided a unified information-theoretic analysis for encoder-generator-type generative models, offering a better understanding of their generalization properties. Our results improved the analysis of VAEs, provided meaningful generalization bounds for DMs, and explicitly unveiled the trade-off on the choice of the diffusion time $T$. Empirical validation on both synthetic and real data verifies our theoretical results. For a discussion of limitations and broader impacts, see Appendix H.

ACKNOWLEDGEMENT

We appreciate constructive feedback from anonymous reviewers and meta-reviewers. Qi Chen is supported by grant number DSI-PDFY3R1P11 from the Data Sciences Institute at the University of Toronto. Florian Shkurti is supported by the Canada First Research Excellence Fund (CFREF).

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

# A  PRELIMINARIES

## A.1  NOTATION TABLE

Table 1: Summary of major notations

| Symbol | Meaning |
|---|---|
| $[m]$ | $\{1, \ldots, m\}$ |
| Upper case letter (e.g. Y) | Random variable |
| Calligraphic letters (e.g. $\mathcal{Y}$) | Support sets of random variables |
| $\mathscr{P}(\mathcal{Y})$ | The set of all the probability measures over $\mathcal{Y}$ |
| $P_Y$ | Marginal distribution of Y |
| $\mathcal{X}$ | Input space |
| $\mathcal{Z}$ | Latent space |
| $\mathcal{F}(\mathcal{X}, \mathcal{Z})$ | $\{f : \mathcal{X} \to \mathcal{Z}\}$, the set of all the measurable functions from $\mathcal{X}$ to $\mathcal{Z}$ |
| $P_Z^f$ | $f \# P_X \in \mathscr{P}(\mathcal{Z})$, the pushforward distribution of $P_X$ through measurable $f$ |
| $P_Z^E$ | $E \# P_X$, the encoded data distribution |
| $P_{Z\|X}$ | The conditional distribution $Z$ given $X$, for forward Markov chain $X \to Z$ |
| $Q_{X\|Z}$ | The conditional distribution $X$ given $Z$ for reverse Markov chain $Z \to X$ |
| $Q_Z$ or $\pi$ | Simple and easy-to-sample distribution, typically a Gaussian |
| $Q_X^G$ | $G \# Q_Z$ or $G \# \pi$, the generated data distribution |
| $Q_{\hat{X}_{T-t}}$ | $G_t \# \pi$, the generated distribution at diffusion time t |
| $p_E(z\|x), p_E(z)$ | Probability densities for $E(X)$ and $P_Z^E$, respectively |
| $q_G(x\|z), q_G(x)$ | Probability densities for $G(Z)$ and $Q_X^G$, respectively |
| $E : \mathcal{X} \to \mathscr{P}(\mathcal{Z})$ | Encoder to map a data point $X \sim P_X$ to a conditional distribution over $\mathcal{Z}$ |
| $\mathcal{E} \subset \mathcal{F}(\mathcal{X}, \mathscr{P}(\mathcal{Z}))$ | Encoder hypothesis set |
| $G : \mathcal{Z} \to \mathscr{P}(\mathcal{X})$ | Generator to be learned that when applied to $Q_Z$, matches data distribution $P_X$ |
| $\mathcal{G} \subset \mathcal{F}(\mathcal{Z}, \mathscr{P}(\mathcal{X}))$ | Generator hypothesis set |
| $\hat{P}_X$ | $\frac{1}{m} \sum_{i=1}^m \delta_{X_i}$, the empirical measure with $m$ observations, where $X_i \sim P_X$. |
| $\Delta_G : \mathcal{X} \times \mathcal{Z} \times \mathcal{X} \to \mathbb{R}_0^+$ | Sample difference loss for the encoder-generator path. |
| $\mathcal{L}_{P_X}^\pi(E, G)$ | $\mathbb{E}_{X \sim P_X} \mathbb{E}_{Z \sim \pi} \mathbb{E}_{\hat{X} \sim G(Z)}[\Delta_G(\hat{X}, Z, X)]$, generation error |
| $\mathcal{L}_{\hat{P}_X}(E, G)$ | $\frac{1}{m} \sum_{i=1}^m \mathbb{E}_{Z \sim E(X_i)} \mathbb{E}_{\hat{X} \sim G(Z)} \Delta_G(\hat{X}, Z, X_i)$, empirical reconstruction error |
| $gen_{P_X}^{\Delta_G}(E, G, \pi)$ | $\mathbb{E}_{S \sim P_X^m}[\mathcal{L}_{P_X}^\pi(E, G) - \mathcal{L}_{\hat{P}_X}(E, G)]$, generalization gap for generation error |
| $T$ | Diffusion time length |
| $\{X_t\}_{t \in [0,T]}$ | Forward diffusion process satisfying $dX_t = f(X_t, t)dt + \lambda(t)dW_t, X_0 \sim P_X$ |
| $\{\overleftarrow{X}_t\}_{t \in [0,T]}$ | Ideal backward diffusion process satisfying $\{X_t\}_{t=0}^T = \{\overleftarrow{X}_t\}_{t=0}^T$ |
| $\{\hat{X}_t\}_{t \in [0,T]}$ | Approximated backward diffusion process with generator $G_t, t \in [0, T]$ |
| $f(\cdot, t) : \mathcal{X} \to \mathcal{X}$ | Drift coefficient |
| $\lambda(t) \in \mathbb{R}$ | Diffusion coefficient |
| $\{W_t\}_{t \in [0,T]}$ | Wiener process/Brownian motion |

## A.2 DEFINITIONS

**Definition A.1** ($f$-divergence). *Let $P$ and $Q$ be two probability measures defined on $\mathcal{X}$ with $P \ll Q$. Given a convex function $f : \mathbb{R}_+ \to \mathbb{R} \cup \{\infty\}$ with a continuous extension at $0$ and $f(1) = 0$, we define the $f-$divergence to be:*

$$\mathbb{D}_f(P\|Q) \overset{def}{=} \mathbb{E}_Q\left[f\left(\frac{P}{Q}\right)\right].$$

As particular instantiations, choosing $f(x) = x \log x$ yields the Kullback-Leibler (KL) divergence $\mathbb{D}_{KL}(P\|Q)$ and $f(x) = \frac{1}{2}(x \log x - (x+1)\log(\frac{x+1}{2}))$ yields the Jensen-Shannon (JS) divergence $\mathbb{D}_{JS}(P\|Q)$.

**Definition A.2** (Fisher Divergence). *Let $P$ and $Q$ be two probability measures defined on $\mathcal{X}$, then, we have the fisher divergence:*

$$\mathbb{D}_{Fisher}(P\|Q) \overset{def}{=} \mathbb{E}_{X \sim P}\left[\|\nabla_X \log p(X) - \nabla_X \log q(X)\|_2^2\right],$$

*where $p(x)$ and $q(x)$ are the probability density functions.*

**Definition A.3** (Lipschitz function). *Let $(\mathcal{W}, \|\cdot\|)$ be a normed space. We say a function $f : \mathcal{W} \to \mathbb{R}$ is L-Lipschitz if for all $w_1, w_2 \in \mathcal{W}$, $|f(w_1) - f(w_2)| \le L\|w_1 - w_2\|$.*

**Definition A.4** (Sub-Gaussian). *Define the cumulant generating function(CGF) of random variable $X$ as $\psi_X(\lambda) \overset{def}{=} \log \mathbb{E}[e^{\lambda(X - \mathbb{E}[X])}]$. $X$ is said to be R-sub-Gaussian if*

$$\psi_X(\lambda) \le \frac{\lambda^2 R^2}{2}, \forall \lambda \in \mathbb{R}.$$

Intuitively, a sub-Gaussian random variable demonstrates exponential tail decay at a rate comparable to a Gaussian random variable. Positive number $R$ is its analog for variance, often called the variance proxy. Entailing many common distributions, sub-Gaussianity is a standard assumption on the residuals in the analysis of ordinary least squares (OLS) and more recently, been widely used to provide non-vacuous bounds for deep learning algorithms Negrea et al. (2019).

**Definition A.5** (Mutual Information). *Let $X$ and $Y$ be arbitrary random variables, and $\mathbb{D}_{KL}$ denote the KL divergence. The mutual information between $X$ and $Y$ is defined as:*

$$I(X;Y) = \mathbb{D}_{KL}(P_{X,Y}\|P_X P_Y)$$

**Definition A.6** (Conditional Mutual Information). *Let $X$, $Y$ and $Z$ be arbitrary random variables, The disintegrated mutual information between $X$ and $Y$ given $Z$ is defined as:*

$$I^Z(X;Y) \overset{def}{=} \mathbb{D}_{KL}(P_{X,Y|Z}\|P_{X|Z}P_{Y|Z}).$$

*The corresponding conditional mutual information is defined as:*

$$I(X;Y|Z) \overset{def}{=} \mathbb{E}_Z[I^Z(X;Y)].$$

**Definition A.7** (Coupling). *Let $(\mathcal{X}, \mu)$ and $(\mathcal{Y}, \nu)$ be two probability spaces. Coupling $\mu, \nu$ means constructing two random variables $X$ and $Y$ on some probability space $(\mathcal{Z}, \pi)$, such that $\mathcal{Z} = \mathcal{X} \times \mathcal{Y}$, $(proj_{\mathcal{X}})\#\pi = \mu$ and $(proj_{\mathcal{Y}})\#\pi = \nu$, which means that $\pi$ is the joint measure on $\mathcal{X} \times \mathcal{Y}$ with marginals $\mu, \nu$ on $\mathcal{X}$ and $\mathcal{Y}$ respectively. The couple $(X, Y)$ is called a coupling of $(\mu, \nu)$.*

**Definition A.8** (Wasserstein-$p$ Distance). *Let the two distributions be defined on the same Polish metric space $(\mathcal{X}, \rho)$, where $\rho(\cdot, \cdot)$ is a metric and $p \in [1, +\infty)$, $\Pi(\mu, \nu)$ is the set of all the couplings (see Definition A.7) of $\mu, \nu$. The Wasserstein distance with order $p$ between $\mu$ and $\nu$ is defined as:*

$$\mathbb{D}_{W_p}(\mu\|\nu) \overset{def}{=} \left[\inf_{\pi \in \Pi(\mu,\nu)} \int_{\mathcal{X} \times \mathcal{X}} \rho(x, x')^p d\pi(x, x')\right]^{1/p}.$$

## A.3 USEFUL LEMMAS

**Lemma A.9** (Donsker-Varadhan Representation [Corollary 4.15 (Boucheron et al., 2013)]). *Let $P$ and $Q$ be two probability measures defined on a set $\mathcal{X}$. Let $g : \mathcal{X} \to R$ be a measurable function, and let $\mathbb{E}_{x \sim Q}[\exp g(x)] \le \infty$. Then*

$$\mathbb{D}_{KL}(P\|Q) = \sup_g\{\mathbb{E}_{x \sim P}[g(x)] - \log \mathbb{E}_{x \sim Q}[\exp g(x)]\}.$$

**Lemma A.10** (Decoupling Estimate Xu & Raginsky (2017)). *Consider a pair of random variables $X$ and $Y$ with joint distribution $P_{X,Y}$, let $\tilde{X}$ be an independent copy of $X$, and $\tilde{Y}$ an independent copy of $Y$, such that $P_{\tilde{X},\tilde{Y}} = P_X P_Y$. For arbitrary real-valued function $f : \mathcal{X} \times \mathcal{Y} \to \mathbb{R}$, if $f(\tilde{X}, \tilde{Y})$ is $R$-sub-Gaussian under $P_{\tilde{X},\tilde{Y}}$, then:*

$$|\mathbb{E}[f(X,Y)] - \mathbb{E}[f(\tilde{X}, \tilde{Y})]| \le \sqrt{2R^2 I(X;Y)}.$$

The above lemma has a generalized extension in Bu et al. (2020), which directly assumes conditions on CGF and can cover other assumptions like sub-gamma.

**Lemma A.11** (Girsonov Theorem, c.f. Theorem 8.6.6. of Oksendal (2013)). *If $\hat{B}_s$ is an Itô process solves $d\hat{B}_s = a(\omega, s)ds + dB'_s$ for $\omega \in \Omega$, $0 \le s \le T$, and $\hat{B}_0 = 0$, where $a(\omega, s)$ satisfies $\mathbb{E}\left[\exp\left(\frac{1}{2}\int_0^T a(w,s)^2 ds\right)\right] < \infty$ for each $\omega$, then $\hat{B}_s$ is a Brownian motion with respect to $Q$, where*

$$\frac{dQ}{dP}(\omega) \overset{def}{=} \exp\left(\int_0^T a(\omega, s)dB'_s - \frac{1}{2}\int_0^T \|a(\omega, s)\|_2^2 ds\right).$$

## B  GENERAL OPTIMIZATION OBJECTIVE AND TWO VIEWPOINTS OF DMS

The following is a regular derivation using basic probability knowledge. Similar results can be found in Kingma et al. (2019) with specific parametric notations in VAE. We now give a general version:

$$\begin{aligned}
\mathbb{D}_{KL}(P_X \| G\#Q_Z) &\le \mathbb{D}_{KL}(P_X \| G\#Q_Z) + \mathbb{E}_X \inf_{E\in\mathcal{E}} \left[\mathbb{D}_{KL}(E(X)\|\frac{G(Z)Q_Z}{G\#Q_Z})\right] \\
&\le \inf_{E\in\mathcal{E}} \left[\mathbb{D}_{KL}(P_X\|G\#Q_Z) + \mathbb{E}_X \mathbb{D}_{KL}(E(X)\|\frac{G(Z)Q_Z}{G\#Q_Z})\right] \\
&= \inf_{E\in\mathcal{E}} \left[\int p(x)\log\frac{p(x)}{q_G(x)}dx \right. \\
&\qquad\qquad\qquad \left. + \int p(x)p_E(z|x)\log\frac{p_E(z|x)}{q(z)q_G(x|z)/q_G(x)}dzdx\right] \\
&= \inf_{E\in\mathcal{E}} \left[\int p(x)p_E(z|x)\log\frac{p(x)p_E(z|x)}{q(z)q_G(x|z)}dzdx\right] \\
&= \inf_{E\in\mathcal{E}} \left[\mathbb{D}_{KL}(P_X E(X)\|G(Z)Q_Z)\right].
\end{aligned}$$

### B.1  DECOMPOSITION FOR VAES

The above general objective can be decomposed as:

$$\begin{aligned}
\inf_{E\in\mathcal{E}} \left[\mathbb{D}_{KL}(P_X E(X)\|G(Z)Q_Z)\right] &= \inf_{E\in\mathcal{E}} \left[\mathbb{E}_{X\sim P_X}\mathbb{E}_{Z\sim E(X)}\left[-\log q_G(X|Z)\right]\right. \\
&\qquad\qquad\qquad \left. + \mathbb{E}_{X\sim P_X}\mathbb{D}_{KL}(E(X)\|Q_Z) - h(P_X)\right] \\
&\propto \inf_{E\in\mathcal{E}} \left[\mathbb{E}_{X\sim P_X}\left(\mathbb{E}_{Z\sim E(X)}\left[-\log q_G(X|Z)\right] + \mathbb{D}_{KL}(E(X)\|Q_Z)\right)\right].
\end{aligned}$$

which is the common VAE objective, with the first term being the reconstruction loss and the second term being the distance of the approximated posterior to the predefined prior. Similar results exist in rate-distortion theory, where the first term is interpreted as distortion, and the second is the rate.

### B.2  DMS AS HIERARCHICAL VAES

Some previous work consider DMs as Hierarchical VAEs, such as Huang et al. (2021); Tzen & Raginsky (2019); Kingma et al. (2021). In this setting, we assume that each encoder is a conditional distribution on previous encoder's output and the initial input $X = X_0 \sim P_X$ with $E_t(X_{t-1}) = P_{X_t|X_{t-1},X}$. Similarly, the generator's output is a conditional distribution on previous generator's

output with $G_{T-t+1}(X_t) = Q_{X_{t-1}|X_t}, X_T = Z \sim Q_Z$. Using similar decomposing approach as VAEs gives the variational objective of diffusion models:

$$\mathbb{D}_{KL}(P_X E_{1:T}(X) \| G_{1:T}(Z)Q_Z)$$

$$= \int p(x)p_{E_1}(x_1|x)...p_{E_T}(z|x_{T-1},x) \log \frac{p(x)p_{E_1}(x_1|x)p_{E_2}(x_2|x_1,x)...p_{E_T}(z|x_{T-1},x)}{q(z)q_{G_1}(x_{T-1}|z)q_{G_2}(x_{T-2}|x_{T-1})...q_{G_T}(x|x_1)}dx...dz$$

$$= \mathbb{E}_{x \sim P_X}\left(\mathbb{D}_{KL}(p_{E_T}(z|x)\|q(z)) + \mathbb{E}_{p_{E_1}(x_1|x)}[-\log q_{G_T}(x|x_1)]\right)$$

$$+ \mathbb{E}_{x \sim P_X}\left(\sum_{t=2}^{T}\mathbb{E}_{p_{E_t}(x_t|x)}\mathbb{D}_{KL}(p_{E_t^{-1}}(x_{t-1}|x_t,x)\|q_{G_{T-t+1}}(x_{t-1}|x_t))\right) - h(P_X)$$

$$\propto \mathbb{E}_{X \sim P_X}\left(\mathbb{D}_{KL}(E_T\#...\#E_1(X)\|Q_Z) + \mathbb{E}_{X_1 \sim E_1(X)}[-\log q_{G_T}(X|X_1)]\right)$$

$$+ \mathbb{E}_{X \sim P_X}\left(\sum_{t=2}^{T}\mathbb{E}_{X_t \sim E_t\#...\#E_1(X)}\mathbb{D}_{KL}(P_{X_{t-1}|X_t,X}^{E_t^{-1}}\|G_{T-t+1}(X_t))\right),$$

where $p_{E_t}(x_t|x_{t-1},x) = \frac{p_{E_t^{-1}}(x_{t-1}|x_t,x)p_{E_t}(x_t|x)}{p_{E_{t-1}}(x_{t-1}|x)}$ and the Markov assumption are used. The above objective is the same as Eq. (11) in Kingma et al. (2021). The first term is the prior loss, the second term is the reconstruction loss, and the last term is the diffusion loss. This formulation is much more complex for conducting a generalization analysis than the one introduced in the following. Hence, the theoretical results of diffusion models will focus on the other.

### B.3 DMs as Time-dependent Mappings

The KL divergence between joint distributions can have the following decomposition:

$$\inf_{E \in \mathcal{E}}[\mathbb{D}_{KL}(P_X E(X)\|G(Z)Q_Z)] = \inf_{E \in \mathcal{E}}\mathbb{D}_{KL}(P_Z^E\|Q_Z) + \mathbb{E}_{Z \sim P_Z^E}\mathbb{D}_{KL}(P_{X|Z}^{E^{-1}}\|G(Z)),$$

where $P_{X|Z}^{E^{-1}}$ is the reverse of $E$ satisfying $P_{Z|X}^E P_X = P_Z^E P_{X|Z}^{E^{-1}}$. Considering DMs as time-dependent mappings, we have:

$$\mathbb{D}_{KL}(P_Z^{E_T}\|Q_Z) + \mathbb{E}_{Z \sim P_Z^{E_T}}\mathbb{D}_{KL}(P_{X|Z}^{E_T^{-1}}\|G_T(Z))$$

$$= \mathbb{D}_{KL}(E_T\#P_X\|Q_Z) + \mathbb{E}_{Z \sim E_T\#P_X}\mathbb{D}_{KL}(P_{X|Z}^{E_T^{-1}}\|G_T(Z))$$

$$\leq \mathbb{D}_{KL}(E_T\#P_X\|Q_Z) + \mathbb{E}_{Z \sim E_T\#P_X}\mathbb{D}_{KL}(P_{X_{1:T}|Z}^{E_{1:T}^{-1}}\|Q_{X_{1:T}|Z}^{G_{1:T}}),$$

where the last step holds by relaxing the last state measure to the path measure under data processing inequality. Then, Song et al. (2020b) upper bound the second term with the weighted score matching objective using Girsanov theorem. Hence, we have shown that VAEs and DMs are inherently optimizing the same objective.

## C Proof of Main Theorem

### C.1 Generalization Bound for Generation (Proof of Theorem 4.1)

**Theorem C.1.** *For any encoder $E \in \mathcal{E}$ and generator $G \in \mathcal{G}$ learned from the training data $S = \{X_i\}_{i=1}^m$, assume that the loss $\Delta_G(\tilde{X}, \tilde{Z}, X)$ is $R$-sub-Gaussian (See definition in Def. A.4) under $P_{\hat{X}, \tilde{Z}, X} = Q_{\hat{X}|Z} \times Q_Z \times P_X$, where $Z \sim Q_Z = \pi, \hat{X} \sim G(Z)$, $\tilde{X}, \tilde{Z}$ are respective independent copy of $\hat{X}$ and $Z$ such that $\tilde{X}, \tilde{Z} \perp\!\!\!\perp X$. Then, $\forall X_i \in S, Z_i \sim E(X_i), \hat{X}_i \sim G(Z_i)$, the generalization gap admits:*

$$|gen_{P_X}^{\Delta_G}(E, G, \pi)| \leq \frac{\sqrt{2}R}{m}\sum_{i=1}^m \sqrt{\mathbb{E}_{X_i}[\mathbb{D}_{KL}(E(X_i)\|\pi)] + I(\hat{X}_i; X_i|Z_i)}.$$

*Proof.* In this case, $\tilde{\hat{X}}, \tilde{Z} \sim Q_{\hat{X}|Z} \times Q_Z$, which satisfies $\tilde{\hat{X}}, \tilde{Z} \perp\!\!\!\perp X$.

For any $\eta \in \mathbb{R}$, let the cumulant generating function (CGF) be

$$\psi_{\tilde{\hat{X}}, \tilde{Z}, X}(\eta) \stackrel{\text{def}}{=} \log \mathbb{E}_{\tilde{\hat{X}}, \tilde{Z}, X}\left[e^{\eta(\Delta_G(\tilde{\hat{X}}, \tilde{Z}, X) - \mathbb{E}[\Delta_G(\tilde{\hat{X}}, \tilde{Z}, X)])}\right]$$

$$= \log \mathbb{E}_{\tilde{\hat{X}}, \tilde{Z}, X}\left[e^{\eta \Delta_G(\tilde{\hat{X}}, \tilde{Z}, X)}\right] - \eta \mathbb{E}_{\tilde{\hat{X}}, \tilde{Z}, X}[\Delta_G(\tilde{\hat{X}}, \tilde{Z}, X)].$$

Let $P_{\hat{X}, Z, X}$ be the joint distribution of $X \sim P_X, Z \sim E(X), \hat{X} \sim G(Z)$. Using the Donsker-Varadhan Representation in Lemma A.9, we have $\forall \eta \in \mathbb{R}$:

$$\mathbb{D}_{KL}(P_{\hat{X}, Z, X} \| P_{\tilde{\hat{X}}, \tilde{Z}, X}) = \mathbb{D}_{KL}(P_{\hat{X}, Z, X} \| Q_{\hat{X}|Z} Q_Z P_X)$$

$$= \sup_{g}\left[\mathbb{E}_{\hat{X}, Z, X} g(\hat{X}, Z, X) - \log \mathbb{E}_{\tilde{\hat{X}}, \tilde{Z}, X}[e^{g(\tilde{\hat{X}}, \tilde{Z}, X)}]\right]$$

$$\geq \eta \mathbb{E}_{\hat{X}, Z, X}[\Delta_G(\hat{X}, Z, X)] - \eta \mathbb{E}_{\tilde{\hat{X}}, \tilde{Z}, X}[\Delta_G(\tilde{\hat{X}}, \tilde{Z}, X)] - \psi_{\tilde{\hat{X}}, \tilde{Z}, X}(\eta).$$
(6)

In addition, the generation error admits

$$\mathcal{L}_{P_X}^{\pi}(E, G) = \mathbb{E}_{X \sim P_X} \mathbb{E}_{Z \sim \pi} \mathbb{E}_{\hat{X} \sim G(Z)}[\Delta_G(\hat{X}, Z, X)],$$

thus, we have:

$$\mathbb{E}_S \mathcal{L}_{P_X}^{\pi}(E, G) = \mathbb{E}_{X \sim P_X} \mathbb{E}_{Z \sim \pi} \mathbb{E}_S \mathbb{E}_{\hat{X} \sim G(Z)}[\Delta_G(\hat{X}, Z, X)]$$

$$= \mathbb{E}_{X \sim P_X} \mathbb{E}_{Z \sim \pi} \mathbb{E}_{\tilde{\hat{X}} \sim Q_{\hat{X}|Z}}[\Delta_G(\tilde{\hat{X}}, Z, X)]$$

$$= \mathbb{E}_{\tilde{\hat{X}}, \tilde{Z}, X}[\Delta_G(\tilde{\hat{X}}, \tilde{Z}, X)],$$

where the first equality holds because $\tilde{\hat{X}}$ is an independant copy of $\hat{X}$.

For the empirical reconstruction error

$$\mathcal{L}_{\hat{P}_X}(E, G) = \mathbb{E}_{X \sim \hat{P}_X} \mathbb{E}_{Z \sim E(X)} \mathbb{E}_{\hat{X} \sim G(Z)}[\Delta_G(\hat{X}, Z, X)]$$

$$= \frac{1}{m} \sum_{i=1}^{m} \mathbb{E}_{Z_i \sim E(X_i)} \mathbb{E}_{\hat{X}_i \sim G(Z_i)} \Delta_G(\hat{X}_i, Z_i, X_i),$$

we have

$$\mathbb{E}_S \mathcal{L}_{\hat{P}_X}(E, G) = \frac{1}{m} \sum_{i=1}^{m} \mathbb{E}_{X_i \sim P_X} \mathbb{E}_{Z_i \sim E(X_i)} \mathbb{E}_{\hat{X}_i \sim G(Z_i)} \Delta_G(\hat{X}_i, Z_i, X_i).$$

Hence, the generalization gap for generation error is

$$gen_{P_X}^{\Delta_G}(E, G, \pi) = \mathbb{E}_S\left[\mathcal{L}_{P_X}^{\pi}(E, G) - \mathcal{L}_{\hat{P}_X}(E, G)\right]$$

$$= \frac{1}{m} \sum_{i=1}^{m}\left(\mathbb{E}_{\tilde{\hat{X}}_i, \tilde{Z}_i, X_i}[\Delta_G(\tilde{\hat{X}}, \tilde{Z}, X_i)] - \mathbb{E}_{\hat{X}, Z, X_i} \Delta_G(\hat{X}_i, Z_i, X_i)\right).$$

Combining this with Eq. (6) gives

$$-\eta gen_{P_X}^{\Delta_G}(E, G, \pi) \leq \frac{1}{m} \sum_{i=1}^{m}\left(\mathbb{D}_{KL}(P_{\hat{X}_i, Z_i, X_i} \| Q_{\hat{X}|Z} Q_Z P_{X_i}) + \psi_{\tilde{\hat{X}}, \tilde{Z}, X_i}(\eta)\right)$$

$$\leq \frac{1}{m} \sum_{i=1}^{m}\left(\mathbb{E}_{X_i} \mathbb{D}_{KL}(P_{\hat{X}_i, Z_i | X_i} \| Q_{\hat{X}|Z} \times \pi) + \psi_{\tilde{\hat{X}}, \tilde{Z}, X_i}(\eta)\right)$$

$$\leq \frac{1}{m} \sum_{i=1}^{m} \mathbb{E}_{X_i} \mathbb{D}_{KL}(P_{\hat{X}_i, Z_i | X_i} \| Q_{\hat{X}|Z} \times \pi) + \frac{\eta^2 R^2}{2}, \forall \eta \in \mathbb{R}$$

$$= \frac{1}{m} \sum_{i=1}^{m}\left(\mathbb{E}_{X_i}[\mathbb{D}_{KL}(E(X_i) \| \pi)] + I(\hat{X}_i; X_i | Z_i)\right) + \frac{\eta^2 R^2}{2}, \forall \eta \in \mathbb{R}.$$

The last inequality is by the $R$-sub-Gaussian assumption and the last equality holds because the reconstruction process and the generation process use the same generator $G$. Dividing both sides by $\eta$, for $\eta > 0$, it gives:

$$-gen_{P_X}^{\Delta_G}(E, G, \pi) \le \frac{1}{m} \sum_{i=1}^{m} \left( \frac{\mathbb{E}_{X_i}[\mathbb{D}_{KL}(E(X_i)\|\pi)] + I(\hat{X}_i; X_i|Z_i)}{\eta} + \frac{\eta R^2}{2} \right), \forall \eta > 0,$$

since $\frac{a}{\eta} + b\eta \ge \sqrt{2ab}, \forall a, b \ge 0, \lambda > 0$, so we have

$$-gen_{P_X}^{\Delta_G}(E, G, \pi) \le \frac{1}{m} \sum_{i=1}^{m} \sqrt{2}R\sqrt{\mathbb{E}_{X_i}[\mathbb{D}_{KL}(E(X_i)\|\pi)] + I(\hat{X}_i; X_i|Z_i)}.$$

Analogously, for $\eta < 0$, we have

$$gen_{P_X}^{\Delta_G}(E, G, \pi) \le \frac{1}{m} \sum_{i=1}^{m} \left( \frac{\mathbb{E}_{X_i}[\mathbb{D}_{KL}(E(X_i)\|\pi)] + I(\hat{X}_i; X_i|Z_i)}{-\eta} + \frac{-\eta R^2}{2} \right), \forall \eta < 0,$$

where we have $P_{\hat{X}_i, Z_i, X_i} = P_{\hat{X}, Z, X}$ because $Z \sim E(X_i), \hat{X} \sim G(Z), X_i \sim P_X$. Hence, it gives

$$gen_{P_X}^{\Delta_G}(E, G, \pi) \le \frac{1}{m} \sum_{i=1}^{m} \sqrt{2}R\sqrt{\mathbb{E}_{X_i}[\mathbb{D}_{KL}(E(X_i)\|\pi)] + I(\hat{X}_i; X_i|Z_i)}.$$

Finally, we get

$$|gen_{P_X}^{\Delta_G}(E, G, \pi)| \le \frac{\sqrt{2}R}{m} \sum_{i=1}^{m} \sqrt{\mathbb{E}_{X_i}[\mathbb{D}_{KL}(E(X_i)\|\pi)] + I(\hat{X}_i; X_i|Z_i)}$$

Concludes the proof. □

## C.2 GENERATION ERROR BOUND (PROOF OF COROLLARY 4.2 AND 4.3)

**Corollary C.2.** *Under Theorem 4.1, let $\Delta_G(\hat{X}, Z, X) = \|\hat{X} - X\|$. Then, the Wasserstein distance between the data distribution $P_X$ and the generated distribution $G\#\pi$ is upper bounded by:*

$$\mathbb{D}_{W_1}(P_X\|G\#\pi) \le \mathbb{E}_S \mathcal{L}_{\hat{P}_X}(E, G) + \frac{\sqrt{2}R}{m} \sum_{i=1}^{m} \sqrt{\mathbb{E}_{X_i}[\mathbb{D}_{KL}(E(X_i)\|\pi)] + I(\hat{X}_i; X_i|Z_i)}.$$

*Proof.* By definition of Wasserstein distance, we have:

$$
\begin{aligned}
\mathbb{D}_{W_1}(P_X\|G\#\pi) &= \inf_{\gamma \in \Pi(P_X, G\#\pi)} \int_{\mathcal{X} \times \mathcal{X}} \|x - x'\| d\gamma(x, x') \\
&\le \mathbb{E}_{X \sim P_X} \mathbb{E}_{\hat{X} \sim G\#\pi} \|\hat{X} - X\| \\
&= \mathbb{E}_{X \sim P_X} \mathbb{E}_{Z \sim \pi} \mathbb{E}_{\hat{X} \sim G(Z)} \|\hat{X} - X\| \\
&= \mathcal{L}_{P_X}^{\pi}(E, G) \\
&\le \mathbb{E}_S \mathcal{L}_{\hat{P}_X}(E, G) + \frac{\sqrt{2}R}{m} \sum_{i=1}^{m} \sqrt{\mathbb{E}_{X_i}[\mathbb{D}_{KL}(E(X_i)\|\pi)] + I(\hat{X}_i; X_i|Z_i)}.
\end{aligned}
$$

The last inequality follows from Theorem 4.1. □

**Corollary C.3.** *Under Theorem 4.1, let the density function of probabilistic decoder $G$ given a latent code $z$ be $q_G(\cdot|z): \mathcal{X} \to \mathbb{R}_0^+$ and $\Delta_G(\hat{X}, Z, X) = -\log q_G(X|Z)$. The KL-divergence between the data distribution $P_X$ and the generated distribution $G\#\pi$ is then upper bounded by:*

$$\mathbb{D}_{KL}(P_X\|G\#\pi) \leq \mathbb{E}_S \mathcal{L}_{\hat{P}_X}(E, G) + \frac{\sqrt{2}R}{m} \sum_{i=1}^m \sqrt{\mathbb{E}_{X_i}[\mathbb{D}_{KL}(E(X_i)\|\pi)] + I(\hat{X}_i; X_i|Z_i)} - h(P_X),$$

*where $h(P_X) = \mathbb{E}_X[-\log p(X)]$ denotes the entropy.*

*Proof.* We have

$$\mathbb{D}_{KL}(P_X\|G\#\pi) = \int p(x) \log \frac{p(x)}{\int q(z)q_G(x|z)dz} dx$$

$$= -h(P_X) - \int p(x) \log \left( \int q(z)q_G(x|z)dz \right) dx$$

$$\leq -h(P_X) - \int p(x)q(z) \log q_G(x|z)dzdx$$

$$= -h(P_X) + \int p(x)q(z) \int q_G(\hat{x}|z)\Delta_G(\hat{x}, z, x)d\hat{x}dzdx$$

$$= -h(P_X) + \mathbb{E}_{X\sim P_X}\mathbb{E}_{Z\sim\pi}\mathbb{E}_{\hat{X}\sim G(Z)}\Delta_G(\hat{X}, Z, X)$$

$$= -h(P_X) + \mathcal{L}_{P_X}^\pi(E, G)$$

$$\leq \mathbb{E}_S \mathcal{L}_{\hat{P}_X}(E, G) + \frac{\sqrt{2}R}{m} \sum_{i=1}^m \sqrt{\mathbb{E}_{X_i}[\mathbb{D}_{KL}(E(X_i)\|\pi)] + I(\hat{X}_i; X_i|Z_i)} - h(P_X).$$

The first inequality is from Jensen's inequality and the last inequality by Theorem 4.1. □

# D  DISCUSSION OF THE VAE BOUND

## D.1  DETAILED COMPARISON TO PREVIOUS VAE BOUND

As discussed in Sec.6.5.2 Alquier et al. (2024), the mutual information bound is tighter than the PAC-Bayes bound in expectation. For more concise form of mutual information bound, we transform the "Catoni-style" PAC Bayes bound (Theorem 5.2) in Mbacke et al. (2023) to the expectation bound by integrating over the high-probability guarantee: Then, for any $\lambda > 0$, the following bound holds for any $E_\phi(x) = \mathcal{N}(\mu_\phi(x), diag(\sigma_\phi^2(x)\mathrm{I}_{d_2}))$ and a fixed $G_\theta(z) = \mathcal{N}(\mu_\theta(x), \mathrm{I}_{d_1})$:

$$D_{W_1}(P_X\|Q_{G_\theta}^\pi) \leq \mathbb{E}_S[\frac{1}{m} \sum_{i=1}^m \mathbb{E}_{Z\sim E_\phi(X_i)}\mathbb{E}_{\hat{X}\sim G_\theta(Z)}\|\hat{X} - X_i\|]$$

$$+ \frac{1}{\lambda}\mathbb{E}_S[\sum_{i=1}^m \mathbb{D}_{KL}(E_\phi(X_i)\|\pi)] + \frac{\lambda\Delta^2}{8m} + \frac{K_\theta}{m}\mathbb{E}_S \sum_{i=1}^m \mathbb{D}_{W_2}(E_\phi(X_i)\|\pi),$$

where $\Delta := sup_{x,x'}\|x - x'\|$ is the diameter of the bounded input space, $K_\theta$ is the Lipchitz constant of $\mu_\theta$. Since $\frac{a}{2\lambda} + \frac{b\lambda}{2} \geq \sqrt{ab}, \forall a, b \geq 0, \lambda > 0$, we have

$$D_{W_1}(P_X\|Q_{G_\theta}^\pi) \leq \mathbb{E}_S \mathcal{L}_{\hat{P}_X}(E_\phi, G_\theta) + \sqrt{\frac{\Delta^2}{2m} \sum_{i=1}^m \mathbb{E}_{X_i}\mathbb{D}_{KL}(E_\phi(X_i)\|\pi)}$$

$$+ \frac{K_\theta}{m} \sum_{i=1}^m \mathbb{E}_{X_i}\mathbb{D}_{W_2}(E_\phi(X_i)\|\pi).$$

Both the Wasserstein-2 distance and the KL-divergence control the generalization of the encoder. Specifically, since $\pi = \mathcal{N}(\mathbf{0}, \mathbf{I})$ is Gaussian, we have $\mathbb{D}_{W_2}(E(X_i)\|\pi) \leq \sqrt{2\mathbb{D}_{KL}(E_\phi(X_i)\|\pi)}$ according to the Transportation Cost Inequality. The above bound can be further formulated as:

$$D_{W_1}(P_X\|Q_{G_\theta}^\pi) \leq \mathbb{E}_S \mathcal{L}_{\hat{P}_X}(E_\phi, G_\theta) + (\frac{\Delta}{\sqrt{2}} + \sqrt{2}K_\theta)\sqrt{\frac{1}{m} \sum_{i=1}^m \mathbb{E}_{X_i}\mathbb{D}_{KL}(E_\phi(X_i)\|\pi)}.$$

To be noted, the above bound only holds for specific $G_\theta$, not all $G_\theta$. In contrast, our bound holds for all $G_\theta$, *i.e.*, it considers the generalization of the generator.

$$\mathbb{D}_{W_1}(P_X \| Q_{G_\theta}^\tau) \le \mathbb{E}_S \mathcal{L}_{\hat{P}_X}(E_\phi, G_\theta) + \frac{\sqrt{2}R}{m} \sum_{i=1}^m \sqrt{\mathbb{E}_{X_i}[\mathbb{D}_{KL}(E_\phi(X_i)\|\pi)] + I(\hat{X}_i; X_i | Z_i)}$$

$$\le \mathbb{E}_S \mathcal{L}_{\hat{P}_X}(E_\phi, G_\theta) + \frac{\sqrt{2}R}{m} \sum_{i=1}^m \sqrt{\mathbb{E}_{X_i}[\mathbb{D}_{KL}(E_\phi(X_i)\|\pi)]}$$

$$+ \frac{\sqrt{2}R}{m} \sum_{i=1}^m \sqrt{I(\hat{X}_i; X_i | Z_i)}.$$

The bounded support assumption w.r.t $\| \cdot \|$ implies sub-Gaussian with $R = \frac{\Delta}{2}$ (Duchi, 2016). Hence, we have:

$$\mathbb{D}_{W_1}(P_X \| Q_{G_\theta}^\tau) \le \mathbb{E}_S \mathcal{L}_{\hat{P}_X}(E_\phi, G_\theta) + \frac{\Delta}{\sqrt{2}} \sqrt{\frac{1}{m} \sum_{i=1}^m \mathbb{E}_{X_i}[\mathbb{D}_{KL}(E_\phi(X_i)\|\pi)]}$$

$$+ \frac{\Delta}{\sqrt{2}} \sqrt{\frac{1}{m} \sum_{i=1}^m I(\hat{X}_i; X_i | Z_i)}.$$

Ignoring the additional generalization term of generator, our bound is tighter than previous work without the unnecessary Wasserstein-2 distance or could be considered has a smaller factor without $\sqrt{2}K_\theta$. However, not all sub-Gaussian random variables are bounded, so our assumption is more flexible and valid for unbounded support. As we discussed in Sec.5, minimizing the first two terms in the bound is equivalent to the empirical $\beta$-VAE objective:

$$\mathcal{L}_{VAE}(\phi, \theta) = \mathcal{L}_{\hat{P}_X}(E_\phi, G_\theta) + \beta \frac{1}{m} \sum_{i=1}^m \mathbb{D}_{KL}(E_\phi(X_i)\|\pi),$$

where $\beta$ is the regularization constant.

## D.2   ESTIMATION OF THE CONDITIONAL MUTUAL INFORMATION TERM

To estimate the bound for VAE, the difficulty lies in estimating $\sqrt{\frac{1}{m} \sum_{i=1}^m I(\hat{X}_i; X_i | Z_i)}$, where the other two terms are easy to compute because $E_\phi$ and $G_\theta$ are tractable distributions in VAEs.

To address this, we can bound the conditional mutual information term as

$$\frac{1}{m} \sum_{i=1}^m I(\hat{X}_i; X_i | Z_i) \le \frac{1}{m} \mathbb{E}_S \left( \sum_{i=1}^m \frac{1}{m} \mathbb{E}_{Z_i \sim E_\phi(X_i)} D_{KL}(G_\theta(Z_i) \| \mathbb{E}_S G_\theta(Z_i)) \right)$$

$$\le \frac{1}{m} \mathbb{E}_S \left( \sum_{i=1}^m \frac{1}{m} \mathbb{E}_{Z_i \sim E_\phi(X_i)} D_{KL}(G_\theta(Z_i) \| G_{\tilde{\theta}}(Z_i)) \right),$$

which can be estimated by sampling several times $m$ data points from the dataset, and then calculating the bound by replacing $\mathbb{E}_S G_\theta(Z_i)$ as some data-free prior.

In the following, we show how to prove the above bound step by step. We first prove that

$$I(\hat{X}_i; X_i | Z_i) \le \frac{1}{m} I(\hat{X}_i; S | Z_i), \forall i \in [m].$$

According to the chain rule in mutual infomation, we have

$$I(\hat{X}_i; S | Z_i) = I(\hat{X}_i; X_{1:m} | Z_i) = I(\hat{X}_i; X_1 | Z_i) + \sum_{j=2}^m I(\hat{X}_i; X_j | Z_i, X_{1:j-1}).$$

Moreover, we have

$$I(\hat{X}_i, X_{1:j-1}; X_j | Z_i) = I(\hat{X}_i; X_j | Z_i, X_{1:j-1}) + I(X_j; X_{1:j-1} | Z_i)$$
$$= I(\hat{X}_i; X_j | Z_i) + I(X_{1:j-1}; X_j | Z_i, \hat{X}_i).$$

Since both encoder and generator are learned from dataset $S$, we have the following Markov chains

$$X_j \to Z_i, X_{1:j-1} \to Z_i; Z_i \to \hat{X}_i, X_j \to \hat{X}_i, X_{1:j-1} \to \hat{X}_i, \forall i, j \in [m],$$

which gives $I(X_j; X_{1:j-1} | Z_i) \le I(X_{1:j-1}; X_j | Z_i, \hat{X}_i)$.

This can be derived with the mutual information chain rule:

$$I(X_j; X_{1:j-1} | Z_i) = I(X_j; X_{1:j-1} | \hat{X}_i, Z_i) - I(X_j; X_{1:j-1}; \hat{X}_i | Z_i),$$

where $I(X_j; X_{1:j-1}; \hat{X}_i | Z_i) \ge 0$. Therefore, we have $I(\hat{X}_i; X_j | Z_i, X_{1:j-1}) \ge I(\hat{X}_i; X_j | Z_i)$, and can further obtain

$$I(\hat{X}_i; S | Z_i) = I(\hat{X}_i; X_{1:m} | Z_i)$$
$$\ge \sum_{j=1}^{m} I(\hat{X}_i; X_j | Z_i) = m I(\hat{X}_i; X_i | Z_i), \forall i \in [m].$$

The last equality holds because learning $G_\theta$ with objective of VAE equally depends on each datapoints, known as the symmetry in stability and generalization (Bousquet & Elisseeff, 2002; Bu et al., 2020).

The mutual information itself is hard to estimate since it's distribution dependent. We could use the variational form by introducing an additional conditional distribution $G_{\tilde{\theta}}(Z_i)$, where $\tilde{\theta}$ is some random initialization of the generator network. Then, we have the following:

$$I(\hat{X}_i; S | Z_i) = \int p(z) \int p(\hat{x}, s | z) \log \frac{p(\hat{x} | s, z)}{q(\hat{x} | z)} d\hat{x} ds dz$$
$$= \int p(s) p(z | s) D_{KL}(G_\theta(z) \| \mathbb{E}_S G_\theta(z)) dz ds$$
$$= \mathbb{E}_S \mathbb{E}_{Z_i \sim E_\phi(X_i)} D_{KL}(G_\theta(Z_i) \| \mathbb{E}_S G_\theta(Z_i))$$
$$\le \mathbb{E}_S \mathbb{E}_{Z_i \sim E_\phi(X_i)} D_{KL}(G_\theta(Z_i) \| \mathbb{E}_S G_\theta(Z_i)) + \mathbb{E}_{Z_i \sim \mathbb{E}_S E_\phi(X_i)} D_{KL}(\mathbb{E}_S G_\theta(Z_i) \| G_{\tilde{\theta}}(Z_i))$$
$$= \mathbb{E}_S \mathbb{E}_{Z_i \sim E(X_i)} D_{KL}(G_\theta(Z_i) \| G_{\tilde{\theta}}(Z_i)).$$

By selecting a duplicated decoder network with random initialization as reference, we can estimate the generalization of the generator $G_\theta$ with only the train data.

### D.3 EXAMPLE ANALYSIS OF LINEAR VAE

We analyze simple linear VAE models following Ichikawa & Hukushima (2023; 2024). Let $E_\phi(x) = N(\mu_\phi(x), \sigma^2 I_{d'}))$ with $\mu_\phi(x) = \frac{1}{\sqrt{d}} \phi^T x, \phi = R^{d \times d'}$ and $G_\theta(z) = N(\mu_\theta(z), I_d)$ with $\mu_\theta = \frac{1}{\sqrt{d}} \theta^T z, \theta \in R^{d' \times d}, \pi = N(0, I_{d'}), x \in R^d, z \in R^{d'}$. Combining the estimation for the conditional mutual information term, where a duplicate randomly initialized generator $G_{\tilde{\theta}}(z) = N(\mu_{\tilde{\theta}}(z), I_d)$ is needed. Then, we have the following bound:

$$\sqrt{2} R \sqrt{\frac{1}{2}(\sigma^2 - 1)d' - d' \log(\sigma^2) + \frac{1}{d} \mathbb{E}_S \left( \frac{1}{m} \sum_{i=1}^{m} x_i^\top \phi \phi^\top x_i \right) + \frac{\sigma^2}{2dm} \|\theta - \tilde{\theta}\|^2}.$$

Consider the proportional limit setting with $\alpha = \frac{m}{d} = \Theta(1)$, we have the following two major terms: $\sqrt{\frac{\alpha}{m} \mathbb{E}_S \left( \frac{1}{m} \sum_{i=1}^{m} x_i^\top \phi \phi^\top x_i \right)}$ for encoder and $\sqrt{\frac{\sigma^2 \alpha}{2m^2} \|\theta - \tilde{\theta}\|^2}$ for generator. We left the convergence analysis as future work.

# E  PROOF FOR DIFFUSION MODELS

## E.1  PROOF OF LEMMA 6.1

**Lemma E.1.** *Let $\{X_t\}_{t=0}^T$ be the empirical version of the forward diffusion process defined in Eq. (3), where $X_0 \sim \hat{P}_X$. We assume the existence of the backward process under the regularity conditions outlined in Song et al. (2021) and denote it as $\{\overleftarrow{X}_t\}_{t=0}^T = \{X_t\}_{t=0}^T$ , which results from the reverse-time SDE defined in Eq. (4). Then, the generative backward process $\{\hat{X}\}_{t=0}^T$ is defined in Eq. (5). Let $E_t, E_t^{-1}, G_t, \forall t \in [0,T]$ be their corresponding time-dependent Markov kernels. The density function of any generator $G$, given a latent code $z$, is denoted as $q_G(\cdot|z) : \mathcal{X} \to \mathbb{R}_0^+$, and let $\Delta_G(\hat{X}, Z, X) = -\log q_G(X|Z)$. Then, we have*

$$|\mathcal{L}_{\hat{P}_X}(E_T, G_T) - \mathcal{L}_{\hat{P}_X}(E_T, E_T^{-1})| \le \frac{1}{2}\int_{t=0}^T \lambda^2(t)\mathbb{D}_{Fisher}(\hat{P}_{X_t}\|Q_{\hat{X}_t})dt\,.$$

*Proof.* From the definition of the empirical reconstruction loss, we have

$$\begin{aligned}
\mathcal{L}_{\hat{P}_X}(E_T, E_T^{-1}) &= \mathbb{E}_{X_0\sim\hat{P}_X}\mathbb{E}_{X_T\sim E_T(X_0)}\mathbb{E}_{\overleftarrow{X}_0\sim E_T^{-1}(X_T)}\Delta_{E_T^{-1}}(\overleftarrow{X}_0, X_T, X_0)\\
&= \mathbb{E}_{X_0\sim\hat{P}_X}\mathbb{E}_{X_T\sim E_T(X_0)}\mathbb{E}_{\overleftarrow{X}_0\sim E_T^{-1}(X_T)}\left(-\log q_{E_T^{-1}}(X_0|X_T)\right)\\
&= \mathbb{E}_{X_0\sim\hat{P}_X}\mathbb{E}_{X_T\sim E_T(X_0)}\left(-\log q_{E_T^{-1}}(X_0|X_T)\right)\,.
\end{aligned}$$

Analogously, we also have

$$\begin{aligned}
\mathcal{L}_{\hat{P}_X}(E_T, G_T) &= \mathbb{E}_{X_0\sim\hat{P}_X}\mathbb{E}_{X_T\sim E_T(X_0)}\mathbb{E}_{\hat{X}_0\sim G(X_T)}\Delta_{G_T}(\hat{X}_0, X_T, X_0)\\
&= \mathbb{E}_{X_0\sim\hat{P}_X}\mathbb{E}_{X_T\sim E_T(X_0)}\mathbb{E}_{\hat{X}_0\sim G_T(X_T)}\left(-\log q_{G_T}(X_0|X_T)\right)\\
&= \mathbb{E}_{X_0\sim\hat{P}_X}\mathbb{E}_{X_T\sim E_T(X_0)}\left(-\log q_{G_T}(X_0|X_T)\right)\,.
\end{aligned}$$

These two empirical reconstruction losses aim to compare the corresponding SDEs, both of which start from a random draw from the aggregate posterior induced by the encoder.

The first is the backward process given the empirical data distribution, characterized by $E_t^{-1}, t \in [0,T]$:

$$d\overleftarrow{X}_t = [f(\overleftarrow{X}_t, t) - \lambda(t)^2\nabla\log\hat{p}_t(\overleftarrow{X}_t)]dt + \lambda(t)dW_t, \overleftarrow{X}_T \sim \hat{P}_{X_T}\,.$$

The second is the generating process, characterized by $G_t, t \in [0,T]$:

$$d\hat{X}_t = [f(\hat{X}_t, t) - \lambda(t)^2\nabla\log q_t(\hat{X}_t)]dt + \lambda(t)d\hat{W}_t, \hat{X}_T \sim \hat{P}_{X_T}\,.$$

Therefore,

$$\begin{aligned}
|\mathcal{L}_{\hat{P}_X}(E_T, G_T) - \mathcal{L}_{\hat{P}_X}(E_T, E_T^{-1})| &= \left|\mathbb{E}_{X_0\sim\hat{P}_X}\mathbb{E}_{X_T\sim E_T(X_0)}\log\left(\frac{q_{E_T^{-1}}(X_0|X_T)}{q_{G_T}(X_0|X_T)}\right)\right|\\
&= \left|\mathbb{E}_{X_T\sim\hat{P}_{X_T}}\int q_{E_T^{-1}}(x_0|X_T)\log\left(\frac{q_{E_T^{-1}}(x_0|X_T)}{q_{G_T}(x_0|X_T)}\right)dx_0\right|\\
&= \mathbb{E}_{X_T\sim\hat{P}_{X_T}}\mathbb{D}_{KL}(Q_{\overleftarrow{X}_0|X_T}^{E_T^{-1}}\|Q_{\hat{X}_0|X_T}^{G_T})\,,
\end{aligned}$$

which gives

$$|\mathcal{L}_{\hat{P}_X}(E_T, G_T) - \mathcal{L}_{\hat{P}_X}(E_T, E_T^{-1})|$$

$$\leq \mathbb{E}_{X_T \sim \hat{P}_{X_T}} \mathbb{D}_{KL}(Q_{(\cdot|X_T)}^{E_T^{-1}} \| Q_{(\cdot|X_T)}^{G_T})$$

$$= \mathbb{E}_{X_T \sim \hat{P}_{X_T}} \mathbb{E}_{Q_{E_T^{-1}}(\cdot|X_T)} \left[ \int_0^T \lambda(t)(\nabla \log \hat{p}_t(X_t) - \nabla \log q_t(X_t)) dW_t \right.$$

$$\left. + \frac{1}{2} \int_{t=0}^T \lambda^2(t) \|\nabla \log \hat{p}_t(X_t) - \nabla \log q_t(X_t)\|_2^2 dt \right]$$

$$= \mathbb{E}_{X_T \sim \hat{P}_{X_T}} \mathbb{E}_{Q_{E_T^{-1}}(\cdot|X_T)} \left[ \frac{1}{2} \int_{t=0}^T \lambda^2(t) \|\nabla \log \hat{p}_t(X_t) - \nabla \log q_t(X_t)\|_2^2 dt \right]$$

$$= \frac{1}{2} \int_{t=0}^T \mathbb{E}_{X_t \sim \hat{P}_{X_t}} [\lambda^2(t) \|\nabla \log \hat{p}_t(X_t) - \nabla \log q_t(X_t)\|_2^2] dt$$

$$= \frac{1}{2} \int_{t=0}^T \lambda^2(t) \mathbb{D}_{Fisher}(\hat{P}_{X_t} \| Q_{\hat{X}_t}) dt \,.$$

The first inequality is obtained by applying data processing inequality to the Markov chain, where the KL divergence between the last iterate conditionals is smaller than that of the whole path, similar to the proof of Theorem 1 in Song et al. (2020b). The subsequent equalities are from the Girsanov Theorem (Theorem 8.6.6 in Oksendal (2013)) and the definition of the Fisher divergence. □

### E.2 PROOF OF THEOREM 6.2

**Theorem E.2.** *Under Lemma 6.1, for any SDE encoder $E_t$ and generator $G_t^\theta$ trained via score matching on $S = \{X_i\}_{i=1}^m$, the corresponding outputs at the diffusion time $T$ are $\hat{X}_T \sim E_T(X_i)$, $\hat{X}_0 \sim G_T^\theta(\hat{X}_T)$ for each $X_i \in S$. The KL-divergence between the original data distribution $P_X$ and the generated data distribution $G_T^\theta\#\pi$ at diffusion time $T$ is then upper bounded by:*

$$\mathbb{D}_{KL}(P_X\|G_T^\theta\#\pi) \leq \mathbb{E}_S\bigg(\underbrace{-\frac{1}{m}\sum_{i=1}^m \mathbb{D}_{KL}(E_T(X_i)\|E_T\#\hat{P}_X) + \hat{\mathcal{L}}_{ESM}(\theta, \lambda(\cdot))}_{T_1}\bigg)$$

$$+ \underbrace{\frac{\sqrt{2}R}{m}\sum_{i=1}^m\sqrt{\mathbb{E}_{X_i}[\mathbb{D}_{KL}(E_T(X_i)\|\pi)]}}_{T_2} + \underbrace{\frac{\sqrt{2}R}{m}\sum_{i=1}^m\sqrt{I(\hat{X}_0; X_i|\hat{X}_T)}}_{T_3}.$$

*Proof.* At first, we combine the results of Corollary 4.3 and Lemma 6.1 and obtain:

$$\mathbb{D}_{KL}(P_X\|G_T^\theta\#\pi) \leq \mathbb{E}_S\left(\mathcal{L}_{\hat{P}_X}(E_T, E_T^{-1}) + \hat{\mathcal{L}}_{ESM}(\theta, \lambda(\cdot))\right)$$

$$+ \frac{\sqrt{2}R}{m}\sum_{i=1}^m\sqrt{\mathbb{E}_{X_i}[\mathbb{D}_{KL}(E_T(X_i)\|\pi)] + I(\hat{X}_0; X_i|\hat{X}_T)} - h(P_X)$$

$$\leq \mathbb{E}_S\left(\mathcal{L}_{\hat{P}_X}(E_T, E_T^{-1}) + \hat{\mathcal{L}}_{ESM}(\theta, \lambda(\cdot))\right) - h(P_X)$$

$$+ \frac{\sqrt{2}R}{m}\sum_{i=1}^m\sqrt{\mathbb{E}_{X_i}[\mathbb{D}_{KL}(E_T(X_i)\|\pi)]} + \frac{\sqrt{2}R}{m}\sum_{i=1}^m\sqrt{I(\hat{X}_0; X_i|\hat{X}_T)}.$$

Then, the reconstruction error $\mathcal{L}_{\hat{P}_X}(E_T, E_T^{-1})$ of the reverse SDE can be decomposed as:

$$\mathcal{L}_{\hat{P}_X}(E_T, E_T^{-1}) = \mathbb{E}_{X_0\sim\hat{P}_X}\mathbb{E}_{X_T\sim E_T(X_0)}[-\log p_{E_T^{-1}}(X_0|X_T)]$$

$$= \mathbb{E}_{X_0\sim\hat{P}_X}\mathbb{E}_{X_T\sim E_T(X_0)}[\log\frac{\hat{p}_T(X_T)}{p_{E_T}(X_T|X_0)\hat{p}_0(X_0)}]$$

$$= h(\hat{P}_X) - \frac{1}{m}\sum_{i=1}^m\mathbb{D}_{KL}(E_T(X_i)\|E_T\#\hat{P}_X),$$

which will converge to $h(\hat{P}_X)$ when $T\to\infty$, because of $E_T\#\hat{P}_X\to\pi$ w.r.t any data distribution and $E_T^{-1}\#\hat{P}_{X_T} = \hat{P}_X$. So if $T\to\infty$ and $m\to\infty$ both hold, we have $\mathbb{E}_S\mathcal{L}_{\hat{P}_X}(E_T, E_T^{-1}) - h(P_X)\to 0$.

Considering the normal case, we have

$$\mathbb{E}_S[h(\hat{P}_X)] - h(P_X) \leq 0,$$

because the entropy function $h(p) = -p\log p$ is concave, and we can take the expectation inside by Jensen's inequality (See Verdú (2019)).

Therefore, we have:

$$\mathbb{E}_S\mathcal{L}_{\hat{P}_X}(E_T, E_T^{-1}) \leq \mathbb{E}_S\left(-\frac{1}{m}\sum_{i=1}^m\mathbb{D}_{KL}(E_T(X_i)\|E_T\#\hat{P}_X)\right) + h(P_X)$$

Combine all these, we conclude the proof. □

# F    PROOF OF THEOREM 6.3

**Theorem F.1.** *Let the step size be $\tau = \frac{T}{N}$, where we split $T$ to $N$ discrete times. For any $k \in [N]$, we use the following discrete update for the backward SDE by setting $\epsilon_{t_k} \sim \mathcal{N}(0, \mathbf{I}_d), t_k = T - \tau k$:*

$$\hat{X}_{t_k} = (1 - \frac{\tau}{2}\lambda^2(T - t_{k-1}))\hat{X}_{t_{k-1}} + \tau\lambda^2(T - t_{k-1})s_\theta(X_{t_{k-1}}, T - t_{k-1}) + \sqrt{\tau}\lambda(T - t_{k-1})\epsilon_{t_{k-1}}.$$

*Furthermore, we assume a bounded score $\nabla_x \log \hat{p}_t(x) \leq L, \forall x, t$. Then, we have the*

$$\frac{1}{m}\sum_{i=1}^m I(\hat{X}_0; X_i|\hat{X}_T) \leq \frac{1}{m}I(\hat{X}_0; X_{1:m}|\hat{X}_T) \leq \frac{TL^2\sum_{k=1}^N \lambda^2(\frac{(k-1)T}{N})}{2mN}.$$

*Proof.* At first, let us denote the sequence of generated data as $\hat{X}^{[N]} \stackrel{\text{def}}{=} [\hat{X}_{t_1}, ..., \hat{X}_{t_N}]$, then, we have the following Markov chain:

$$X_{1:m} \to \hat{X}^{[N]} \to \hat{X}_0$$
$$\uparrow$$
$$\hat{X}_T$$

According to the mutual information chain rule, we have:

$$I(\hat{X}_0; X_{1:m}|\hat{X}_T) = I(\hat{X}_0; X_1|\hat{X}_T) + \sum_{i=2}^m I(\hat{X}_0; X_i|\hat{X}_T, X_{1:i-1}) \geq \sum_{i=1}^m I(\hat{X}_0; X_i|\hat{X}_T).$$

Since $I(\hat{X}_0; X_i|\hat{X}_T, X_{1:i-1}) + I(X_i; X_{1:i-1}|\hat{X}_T) = I(\hat{X}_0, X_{1:i-1}; X_i|\hat{X}_T)$ that can also be decomposed as $I(\hat{X}_0; X_i|\hat{X}_T) + I(X_{1:1-i}; X_i|\hat{X}_T, \hat{X}_0)$ and $I(X_i; X_{1:i-1}|\hat{X}_T) = 0$, the last inequality holds with $I(X_{1:1-i}; X_i|\hat{X}_T, \hat{X}_0) \geq 0$.

For any $k \in [N]$, we use the following discrete update for the backward SDE, where $\epsilon_{t_k} \sim \mathcal{N}(0, \mathbf{I}_d), t_k = T - \tau k, \tau = \frac{T}{N}$.

$$\hat{X}_{t_k} = (1 - \frac{\tau}{2}\lambda^2(T - t_{k-1}))\hat{X}_{t_{k-1}} + \tau\lambda^2(T - t_{k-1})s_\theta(X_{t_{k-1}}, T - t_{k-1}) + \sqrt{\tau}\lambda(T - t_{k-1})\epsilon_{t_{k-1}}.$$

Since the approximation $s_\theta(X_{t_{k-1}}, T - t_{k-1}) \approx \nabla_{X_{t_{k-1}}} \log \hat{p}_{T-t_{k-1}}(X_{t_{k-1}})$ is determined by $S = X_{1:m}$ under some functional form, simply denote it as $g_S(X_{t_{k-1}})$ we can consider the above update as a Langevin dynamics

$$\hat{X}_{t_k} = (1 - \frac{\tau}{2}\lambda^2(T - t_{k-1}))\hat{X}_{t_{k-1}} + \eta_{k-1}g_S(X_{t_{k-1}}) + \sigma_{k-1}\epsilon_{t_{k-1}},$$

where $\eta_{k-1} = \tau\lambda^2(T - t_{k-1}) = \sigma_{k-1}^2$. Then, we can apply the technique in Pensia et al. (2018), and obtain:

$$I(\hat{X}_0; X_{1:m}|\hat{X}_T) \leq I(\hat{X}^{[N]}; X_{1:m}|\hat{X}_T) \leq \sum_{k=1}^N I(\hat{X}_{t_k}; X_{1:m}|\hat{X}_T, \hat{X}^{[k-1]})$$

$$= \sum_{k=1}^N \left( h(\hat{X}_{t_k}|\hat{X}_T, \hat{X}^{[k-1]}) - h(\hat{X}_{t_k}|X_{1:m}, \hat{X}_T, \hat{X}^{[k-1]}) \right)$$

According to the Langevin dynamic update and the bounded gradient assumption, we have

$$h(\hat{X}_{t_k}|\hat{X}_T, \hat{X}^{[k-1]}) = h(\eta_{k-1}g_S(X_{t_{k-1}}) + \sigma_{k-1}\epsilon_{t_{k-1}}) \leq \frac{d}{2}\log\left(2\pi e\frac{\eta_{k-1}^2 L^2 + d\sigma_{k-1}^2}{d}\right),$$

where we use the fact that the Gaussian distribution has the largest entropy with $h(Y) \leq \frac{d}{2}\log\left(\frac{2\pi eC}{d}\right)$ for all random variables $Y$ satisfying $\mathbb{E}\|Y\|_2^2 \leq C$. Moreover, we have

$$\mathbb{E}\|\eta_{k-1}g_S(X_{t_{k-1}}) + \sigma_{k-1}\epsilon_{t_{k-1}}\|_2^2 = \mathbb{E}\|\eta_{k-1}g_S(X_{t_{k-1}})\|_2^2 + \mathbb{E}\|\sigma_{k-1}\epsilon_{t_{k-1}}\|_2^2 \leq \eta_{k-1}^2 L^2 + d\sigma_{k-1}^2,$$

which is due to the independence between the score estimation and the injected noise. Then, we also have $h(\hat{X}_{t_k}|X_{1:m}, \hat{X}_T, \hat{X}^{[k-1]}) = h(\sigma_{k-1}\epsilon_{t_{k-1}}) \leq \frac{d}{2}\log\left(2\pi e\sigma_{k-1}^2\right)$.

Combine all these, we can get:

$$I(\hat{X}_0; X_{1:m}|\hat{X}_T) \leq \sum_{i=1}^{N} \frac{d}{2}\log\left(1 + \frac{\eta_{k-1}^2 L^2}{d\sigma_{k-1}^2}\right) \leq \sum_{i=1}^{N} \frac{\eta_{k-1}^2 L^2}{2\sigma_{k-1}^2},$$

where the last inequality use $\log(1+x) \leq x, \forall x \geq 0$. Putting $\eta_{k-1} = \tau\lambda^2(T - t_{k-1}) = \sigma_{k-1}^2$ into the above equation, we conclude the proof. $\qquad\square$

# G    EXPERIMENT DETAILS

In this section, we provide the detailed experimental setting and some additional experimental results. Our experimental code is available at `https://github.com/livreQ/InfoGenAnalysis`. The implementation is based on the code in Van den Burg & Williams (2021) for VAEs and based on the code in Huang et al. (2021) for diffusion models.

**Computational Resource** The experiments for Swill Roll data were running on a machine with 1 2080Ti GPU of 11GB memory. The experiments for MNIST and CIFAR10 were running on several server nodes with 6 CPUs and 1 GPU of 32GB memory.

## G.1    VAE

To verify our theoretical results in Theorem 5.1 and compare to previous PAC Bayes bound for VAEs, we present experiments on MNIST in this section.

### G.1.1    RESULTS

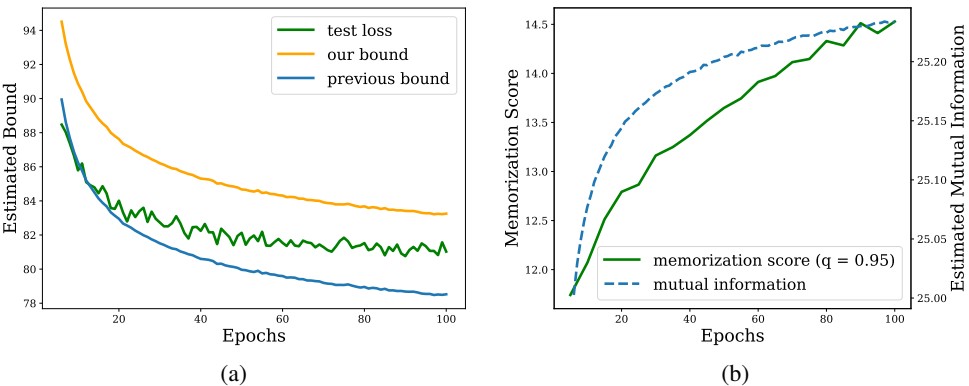

(a)                                    (b)

Figure 4: (a)The test VAE loss, previous PAC Bayes bound (converted to expectation bound) and our mutual information bound on MNIST dataset change over training epochs. (b) The memorization score with $0.95$ quantile and mutual information estimation change over training epochs.

We conducted experiments on the MNIST dataset, using a Bernoulli distribution as the generator $G_\theta$ for the VAE. The mutual information term was estimated using the method proposed in Sec. D.2, leveraging a randomly initialized duplicate generator, which remained fixed as a reference throughout the process.

The estimated bounds and test loss are plotted in Fig.4 (a). The previous PAC Bayes bound estimated with train data goes below the test loss. This is due to the bound does not hold for any $G_\theta$. In contrast, our bound estimated on train data well aligns with the test loss because we have considered the generalization term for $G_\theta$.

To further illustrate the effectiveness of using the conditional mutual information term to capture the generalization of $G_\theta$. We compare its estimation to the memorization score proposed in Van den Burg & Williams (2021), which measures how much more likely an observation is when it is included in the training set than when it is not. The result is presented in Fig.4(b), where we plot the $95\%$ quantile of the memorization score and the conditional mutual information term along the training epochs. The two terms are highly correlated with similar evolving trends, suggesting our bond may capture the memorization to some extent. To be noted, evaluating the memorization score requires an additional validation set, while our bound can be evaluated with only the training set.

### G.1.2 EXPERIMENT SETTINGS

This section strictly follows the setting in Van den Burg & Williams (2021).

**Data and network structure**  During training on MNIST, we dynamically binarize the images by treating each grayscale pixel value as the parameter of an independent Bernoulli random variable, following standard practice. The encoder block is the stack of FC(1024, 512), RELU, FC(512, 256), RELU and FC(256, 16). The generator block is the stack of FC(16, 256), RELU, FC(256, 512), RELU, FC(512, 1024), and Sigmoid.

**Training Details**  We optimized the model parameters using the Adam optimizer with a learning rate of $\eta = 10^{-3}$. The training was conducted with a batch size of 64, while the remaining Adam hyperparameters were kept at their default values in PyTorch. The model was trained for 100 epochs.

### G.2 DIFFUSION MODEL

### G.2.1 BOUND ESTIMATION

Our main objective is to verify Theorem 6.2 for diffusion models, which involves illustrating: 1. the inequality holds, as illustrated in Fig. 2. 2. the evolving trend of the two sides follows the change of sample size $m$, seen in Fig. 2 (a). 3. the trade-off on diffusion time $T$, showed in Fig. 2 (b). To make such a comparison, we need a quantitative estimation of the two sides.

Recall that $\mathbb{D}_{KL}(P_X \| Q_{G_T^\theta}^\pi)$ measures the proximity of the original data distribution to the generated data distribution. A similar metric used to evaluate the performance of generative models is the Fréchet inception distance (FID), which is the Wasserstein-2 distance between the generated and the original data distribution. Since the data distribution is unknown, we conduct a Monte Carlo estimation $\sum_{i=1}^{m_t} \log(p(\tilde{X}_i)/q_{G_\theta}(\tilde{X}_i))$ using a test dataset of size $m_t = 1000$, which is independent of the training set with $S^{te} = \{\tilde{X}\}_{i=1}^{m_t}, \tilde{X}_i \sim P_X$. Then, we use a Kernel Density Estimation (KDE) to calculate both $p(\tilde{X}_i)$ and $q_{G_\theta}(\tilde{X}_i)$. Such estimation of $q_{G_\theta}$ is disentangled from the diffusion process itself and can better reflect the generalization of the learned diffusion model by only using the generated data (one can sample any number of data as wanted to fit the KDE).

On the Right-Hand Side (RHS), $T_2$ has an analytic form. $T_3$ is upper bounded by Theorem 6.3, where we use a step-wise estimation for the maximum score norm $L$ similar to the gradient norm estimation in the literature of information-theoretic learning (Pensia et al., 2018; Li et al., 2019; Negrea et al., 2019). $T_1$ is the KL divergence between the final time posterior and the aggregated posterior, where the former is a multivariate Gaussian, and the latter is a mixture of Gaussian in the normal setting of diffusion models. Thus, we can simply use a KDE with a Gaussian kernel and bandwidth fixed to time-specific variances of the forward process to approximate the aggregated posterior. Combining the empirical score matching loss, we have the estimation of RHS. W.r.t the expectation over $S$, we conduct 5-times Monte-Carlo estimation by randomly generating train datasets with different random seeds.

### G.2.2 SWISS ROLL

**Experimental Setting**

- **Score matching model structure** We use a 4-layer Multilayer Perceptrons (MLPs) with hidden size 128 to approximate the score function, where the input dimension is the 2D data dimension plus the 1D time dimension.
- **Train-test details** During experiments, we record the generated data and estimate their KL divergence from the test data during the training dynamics. We use 1000 Monte Carlo sampling for the test-data KL divergence estimation and the same sampling size for every kernel density estimation. The score matching model $s_\theta(x, t)$ is trained for 10000 iterations, and the backward generation takes 1000 steps, *i.e.*, $N = 1000$.

**Additional Results** In Fig. 5 and Fig. 6, we plot the generated data with the score model obtained at the last iteration for each specific setting, *e.g.*, different train sample size $m$ and diffusion time $T$.

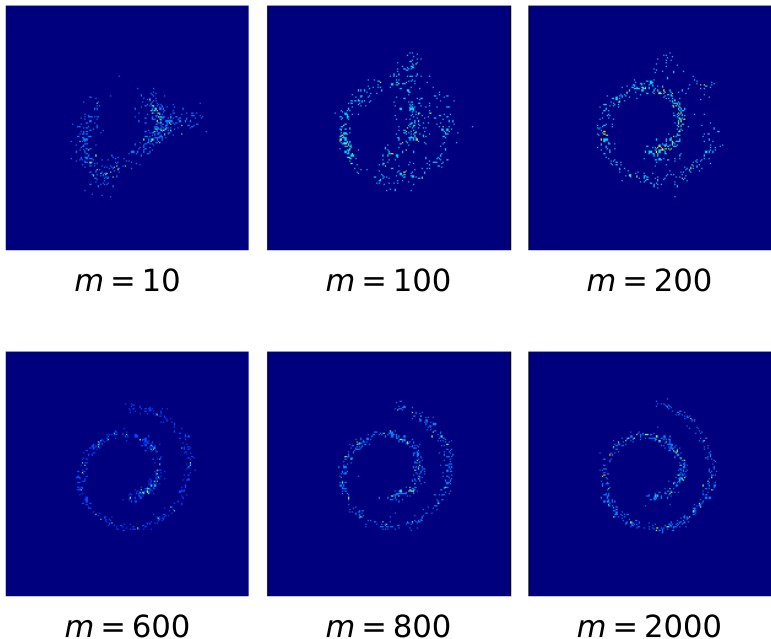

Figure 5: Sampling results w.r.t. different train data size $m$: 1000 data points generated by a score-based model trained with 10000 gradient iterations and diffusion time $T = 1$. The sampling is conducted after 1000 steps when solving the discretized backward SDE.

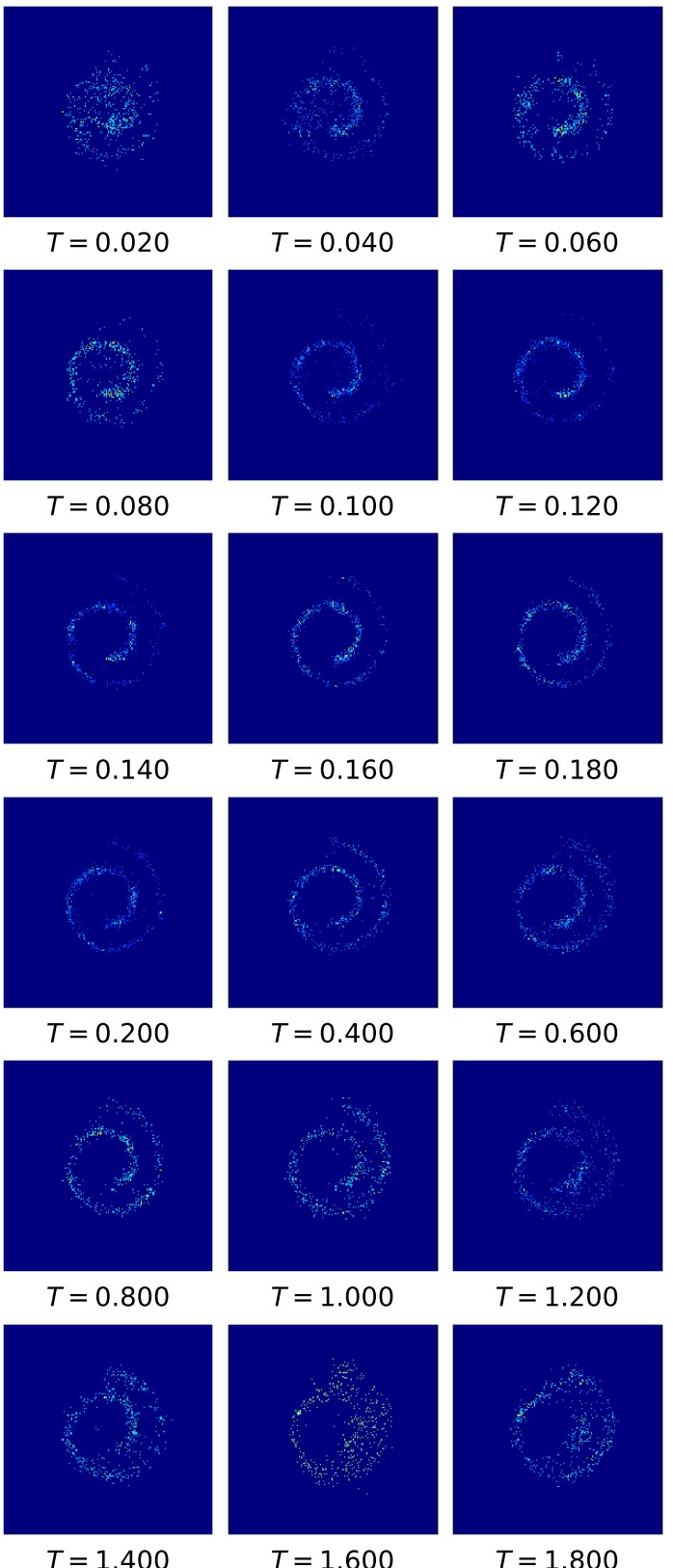

Figure 6: Sampling results w.r.t. different diffusion time $T$: 1000 data points generated by a score-based model trained on $m = 200$ data points with $10000$ gradient iterations.

### G.2.3 MNIST AND CIFAR10

**Experimental Setting**

- **Score matching model structure** Following Ho et al. (2020) and Huang et al. (2021), we use modified UNets based on a Wide ResNet for 1x28x28 images in MNIST data and 3x32x32 images in CIFAR10 data, respectively. The weight normalization is replaced with group normalization. Each model consists of two convolutional residual blocks per resolution level, along with self-attention blocks at the 16x16 resolution between the convolutional layers. The diffusion time $t$ is incorporated into each residual block via the Transformer sinusoidal positional embedding.

- **Train-test details** During experiments, we record the generated data and estimate their KL divergence from the test data during the training dynamics. We use 100 Monte Carlo sampling for test-data KL estimation and the same sampling size for every kernel density estimation. The score matching model $s_\theta(x, t)$ is trained was trained on $m = 16$ images for 10000 iterations, and the backward generation takes 1000 steps, *i.e.*, $N = 1000$. BPD was estimated using the method proposed in Huang et al. (2021), and test data KL was estimated using KDE as for the Swiss Roll data.

**Additional Results**  Large step sizes will cause instability or large discretization errors. However, the step size is not the smaller, the better. After some threshold, reducing the step size further yields negligible improvements because of the model's approximation error and will cause a heavy computation burden. In addition, according to the relation $N = \frac{T}{\tau}$, a small step size corresponds to a large number of steps, which can lead to overfitting, especially when the model is trained with few data.

Replace $N$ with $T/\tau$ in the bound, we have $T_3 = \frac{R\sqrt{(\beta_1 - \beta_0)L^2T^2 + ((1+\tau)\beta_0 - \tau\beta_1)L^2T)}}{\sqrt{m}}$. In the experimental setting of the original submission Fig.7 (a), we set $N = 1000, T \in [0.2, 2]$, so we have $0.0001 \leq \tau \leq 0.002$. Since we used $\beta_0 = 0.1, \beta_1 = 20$ in all the experiments, which gives $0.06 \leq (1 + \tau)\beta_0 - \tau\beta_1 \leq 0.09998$. Therefore, we have $T_3 \in \mathcal{O}(T)$ that has a linear growth w.r.t $T$ for all $\tau$ used in our experiments. Hence, we suppose the impact of $\tau$ in this range is minor. To further verify this, we set $\tau = 0.001$ as suggested by the reviewer and change $N$ for different $T$ accordingly. In Fig.7 (b), we compare the results with the previous setting. It shows the whole upper bound (including score matching loss), and the log density remains consistent with the results in the previous setting. However, we keep using $\tau = 0.001$ for the rest of the experiments to avoid potential concerns because we are varying $T$ in a larger range.

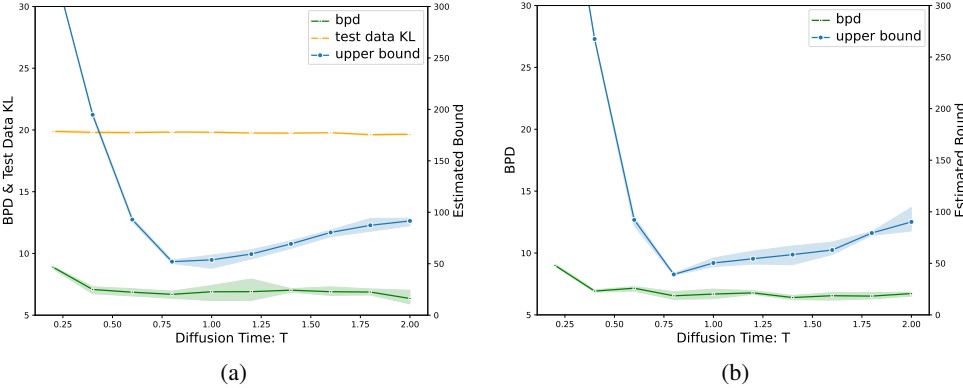

Figure 7: The evolution of the estimated bounds, test data KL divergences, and test data log densities (measured by BPD) w.r.t different diffusion time $T$ for DM trained on few-shot MNIST data ($m = 16$): (a) with fixed number of steps $N = 1000$ (KL and BDP were calculated with 100 test samples,) and (b) with fixed step size $\tau = 0.001$, BDP was calculated with 10000 test samples, note $N = \frac{T}{\tau}$.

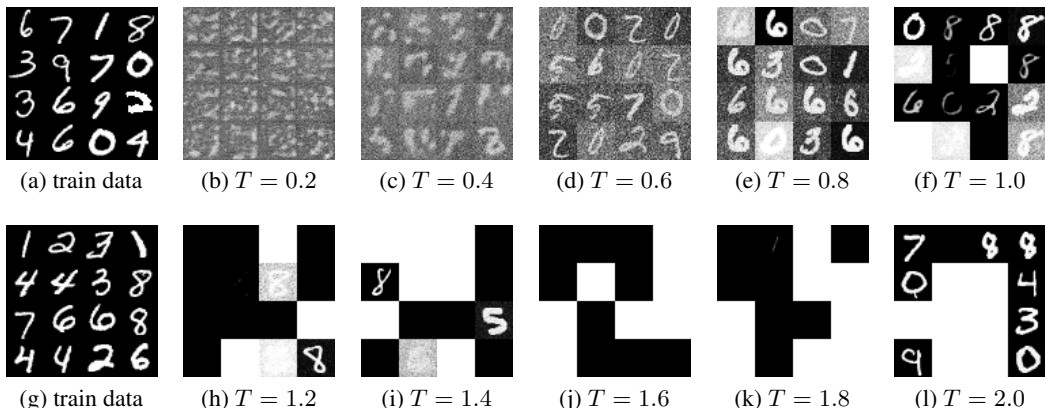

Figure 8: The generated images for different diffusion times $T$ on the MNIST dataset (we randomly sample 16 images for each $T$). The score-matching model was trained on $m = 16$ images (we use this few-shot setting to make sure we can present visual difference within limited random draws) randomly sampled from the dataset for each $T$. The training process takes 10000 iterations. The generation quality is consistent with the estimated bound in Fig. 2 (a), where the optimal diffusion time should be around $T = 0.8$. The sampling process has a fiexed step size $\tau = 0.001$.

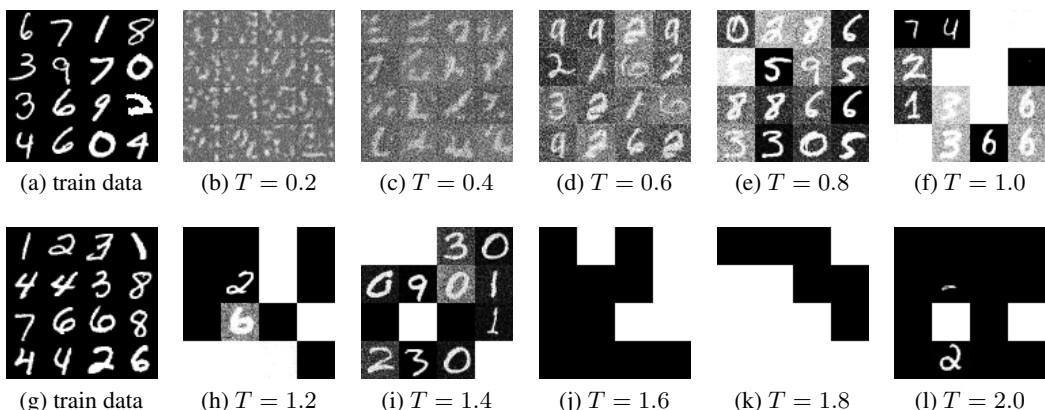

Figure 9: The generated images for different diffusion times $T$ on the MNIST dataset (we randomly sample 16 images for each $T$). The score-matching model was trained on $m = 16$ images (we use this few-shot setting to make sure we can present visual difference within limited random draws) randomly sampled from the dataset for each $T$. The training process takes 10000 iterations. The generation quality is consistent with the estimated bound in Fig. 2 (a), where the optimal diffusion time should be around $T = 0.8$. The sampling process has a fixed number of steps $N = 1000$.

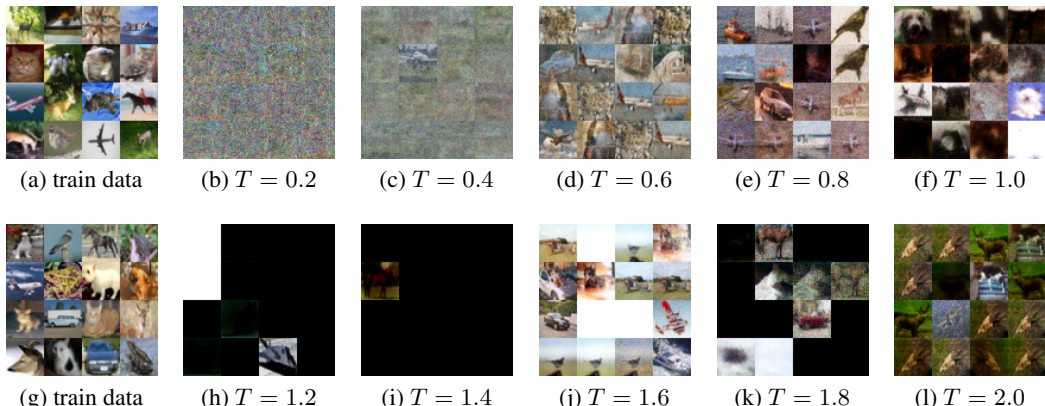

| (a) train data | (b) $T = 0.2$ | (c) $T = 0.4$ | (d) $T = 0.6$ | (e) $T = 0.8$ | (f) $T = 1.0$ |
| (g) train data | (h) $T = 1.2$ | (i) $T = 1.4$ | (j) $T = 1.6$ | (k) $T = 1.8$ | (l) $T = 2.0$ |

Figure 10: The generated images for different diffusion times $T$ on the CIFAR10 dataset (we randomly sample 16 images for each). The score-matching model was trained on $m = 16$ images (we use this few-shot setting to make sure we can present visual difference within limited random draws) randomly sampled from the dataset for each $T$. The training process takes 10000 iterations. The generation quality is consistent with the estimated bound in Fig. 2 (b), where the optimal diffusion time should be around $T = 0.8$. The sampling process has a fixed number of steps $N = 1000$.

## H  BROADER IMPACTS AND LIMITATIONS

**Broader Impacts**

- **Potential positive impacts** We study the theoretical aspects of generative models. By improving the generalization, we can decrease replicated generation to help address the privacy and copyright issues in generative models.

- **Potential negative impacts** The improvement for generating diverse data may be used to generate harmful information or fake news.

- **How to address the potential negative impacts?** We can design harmful information detection mechanisms and embed a filter strategy for generative models.

**Limitations**    The theoretical analysis for DM in the last theorem is for a first-order Euler-Maruyama solver for SDE. One can extend and prove guarantees for other backward SDEs. While some diffusion models use deterministic encoders or generators that also work well in practice (*e.g.*, Song et al. (2020a) and Bansal et al. (2024)), we consider the encoder and generator as randomized mappings as is most typical in diffusion models. However, our current definition of encoder and generator with randomized mapping covers the deterministic mapping setting by restricting $E(X)$ and $G(Z)$ to the set of delta distributions. The problem is due to the mutual information terms could be infinite for deterministic settings. However, this could be addressed by exploiting other refined information-theoretic tools. As this paper focuses on providing a unified theoretical viewpoint for typical VAEs and DMs, we cannot cover all the methods. The improvements mentioned above will be left as future work. Moreover, potential fairness issues raised in learning generative models could be considered by exploiting previous related works like Xu et al. (2024); Shui et al. (2022).

## I  ADDITIONAL RELATED WORKS

**Algorithms for VAEs**    The VAE (Kingma & Welling, 2013) has been widely applied and improved algorithmically through numerous extensions that include changing the posterior distribution to exponential families (Shi et al., 2020; Shekhovtsov et al., 2021) and location-scale families (Park et al., 2019), balancing the rate-distortion trade-off (Higgins et al., 2017; Rybkin et al., 2021), replacing the regularization term with adversarial objectives (Makhzani et al., 2015), or using other divergences like the Wasserstein distance (Tolstikhin et al., 2017) (WAE).

**Score-based diffusion models**    Song et al. (2020b) unifies the previous two main diffusion approaches: Score matching with Langevin dynamics (SMLD) (Song & Ermon, 2019) and Diffusion probabilistic modeling (DDPM) (Sohl-Dickstein et al., 2015; Ho et al., 2020) as score-based diffusion models, where their forward processes are considered as different families of Stochastic Differential Equations (SDEs). Later on, the variational perspective of these models was studied in (Huang et al., 2021; Kingma et al., 2021; Franzese et al., 2023). Huang et al. (2021); Song et al. (2020b) study how to use diffusion models to estimate the data likelihood based on some theoretical results in stochastic calculus (Karatzas & Shreve, 2014; Oksendal, 2013). Recently, latent diffusion models (Vahdat et al., 2021; Rombach et al., 2022) have gained great success in generating high-resolution images, extended to further applications like text-to-image editing (Han et al., 2024; Huberman-Spiegelglas et al., 2024).

**Convergence theory for diffusion models**    De Bortoli et al. (2021) are the first to give quantitative convergence results for DMs, where they upper bound the original and generated data distribution in Total Variation (TV) distance and assume a $L^\infty$-accurate score estimation. This leads to vacuous results under the manifold assumption, where the TV can be very large, even if the distributions are similar. Lee et al. (2023); Chen et al. (2022) that also bound in TV but assume $L^2$-accurate score estimation have the same problem. Chen et al. (2023a) provide an improved analysis with minimal smoothness assumptions, which is valid for any data distribution with second-order moment with a $L^2$-accurate score estimation. The above works focus on analysis for convergence w.r.t population data distribution without explicit consideration of generalization.

**Information-theoretic learning theory**    Recent information-theoretic analyses (Xu & Raginsky, 2017; Russo & Zou, 2019; Steinke & Zakynthinou, 2020; Haghifam et al., 2021; 2022; Hellström & Durisi, 2022; Wang & Mao, 2023) have provided a rigorous framework for understanding the generalization capabilities of deep learning models, and have further been extended to complex learning scenarios, such as meta-learning (Chen et al., 2021; 2023b) and domain adaptation (Wu et al., 2022; Chen & Marchand, 2023). Unlike conventional VC-dimension and uniform stability bounds, this approach offers a key advantage in capturing dependencies on the data distribution, hypothesis space, and learning algorithm.

