# OpenReview forum: "Generalization in VAE and Diffusion Models: A Unified Information-Theoretic Analysis"
_ICLR.cc/2025/Conference — ICLR 2025 Poster_

### Official Review · Reviewer_o393 · 2024-10-24

**Soundness:** 2
**Presentation:** 3
**Contribution:** 3
**Rating:** 6
**Confidence:** 4

**Summary:**

This work puts a bound on the generalization gap for generative models relying on an encoder-decoder architecture.
Using this, they introduce bounds on
 - the wasserstein distance between the data distribution and generated distribution for variational autoencoders (VAEs),
 - the KL-divergence between the data distribution and generated distribution for diffusion models.

They run experiments on a score based diffusion model using data from "Swiss Roll", MNIST and CIFAR-10.

**Strengths:**

S1: VAEs and diffusion models are widely used, so theoretical work on them is very relevant.

S2: The bounds introduced in the work are new.

S3: The story of the article is easy to follow.

**Weaknesses:**

The weaknesses relate to the experimental part of the paper.

W1: For both experiments you are using a fixed number of steps for generation while changing T.
	This means that for varying T, you are also varying the step-size, which can have a large effect
	on generation (See e.g. Zhang and Chen 2023 "Fast Sampling of Diffusion Models with Exponential Integrator").
	It is not clear to me whether this also affects your experiment in other ways (you should check this),
	but if you want to say something about sample quality, my suggested solution is to choose a step size, e.g. 1/1000,
	and also generate images with varying T, but fixed step-size.

W2: Figure 3 b) and c) do not support your claim. You say that the discrepancy comes from using too few samples
	to get a reliable estimation of KL divergence. My suggested solution is to do the experiment with enough samples
	to get a reliable estimation. Since your result is about KL-Divergence and not sample quality, it is not enough to
	look at sample quality for verification. Especially since since performance in one is not directly linked to
	performance in the other. This observation is also made in Theis et al. 2015 "A note on the evaluation of generative models"
	which you also cite yourselves.

**Questions:**

Q1: Do you have a theoretical reason (or at least an intuition) for why the mutual information should follow
the upper bound in theorem 6.3?

Q2: At the end of section 6.1, you write "$T_3$, which characterizes the generalization of generator $G_T$ , will
remain non-zero for a small sample size". How small does the sample size need to be? And if you use enough samples in your
MNIST and CIFAR-10 experiments to get a good estimation of KL-Divergence, will that make the contribution of $T_3$
very small?

Q3: Could you add the next steps in your proof in appendix 1.C? As it is now, I don't see the final claim of the theorem
in the last lines of the proof.

---

> ### Author Response · Authors · 2024-11-24
> **Part 1 --- Weakness 1**
>
> We sincerely thank the reviewer for the valuable and constructive feedback. Below, we address your concerns in detail.
>
> ---
> > **W1. Fixed number of steps Vs. fixed step size.**
>
> Thanks for the insightful question and introducing the interesting paper [1].
>
> * We agree that large step sizes will cause instability or large discretization errors. However, the step size is not the smaller, the better. After some threshold, reducing the step size further yields negligible improvements because of the model's approximation error and will cause a heavy computation burden. In addition, according to the relation $N = \frac{T}{\tau}$, a small step size corresponds to a large number of steps, which can also lead to overfitting, especially when the model is trained with few data.
>
> * Theoretically, varying steps **do not** affect our previous experiments.
>     - Putting $N = \frac{T}{\tau}$ into Theorem 6.3 gives $\frac{TL^2\sum_{k=1}^N \lambda^2(\frac{(k-1)T}{N})}{2mN} = \frac{\tau L^2 \sum_{k=1}^N (\beta_0 + (\beta_1 -\beta_0) (k-1)\tau)}{2m} = \frac{\beta_0 T L^2 + (\beta_1-\beta_0)L^2(T^2 - T\tau)}{2m}$.
>     - Combining Theorem 6.2, we have $T_3 = \frac{R \sqrt{(\beta_1 - \beta_0)L^2T^2 + ((1+\tau)\beta_0 - \tau\beta_1)L^2 T)}}{\sqrt{m}} $.
>
>     - In the experimental setting of the original submission, we have set $N=1000, T \in [0.2, 2]$, so we have $0.0002\leq\tau\leq 0.002$. We have used $\beta_0=0.1, \beta_1=20$ in all the experiments, which gives $0.06\leq(1+\tau)\beta_0 - \tau\beta_1 \leq 0.09998$. Therefore we have $T_3 \in O(T)$ that has a linear growth w.r.t $T$ for all $\tau$ used in our experiments. Hence, we suppose the impact of $\tau$ in this range is minor.
> * Empirically, varying steps also do **not affect** our previous experiments .
>     - To further verify this, we set $\tau=0.001$ as suggested by the reviewer and change $N$ for different $T$ accordingly. In **Fig.7** in the Appendix, we compare the results with the previous setting. It shows the whole upper bound (including score matching loss), and the log density remains consistent with the results in the previous setting.
>
> However, we keep using $\tau=0.001$ for all additional experiments to avoid potential concerns because we will vary T in a larger range.

---

> > ### Author Response · Authors · 2024-11-24
> > **Part 2 --- Weakness 2 & Question 1, 2, 3**
> >
> > ---
> > > **W2. Figure 3 (b) and (c) do not sufficiently support the claim.**
> >
> > Thanks for the suggestion. We do agree that test on large dataset can help verify the trade-off.
> >
> > * Additional experiments using more data verify the trade-off
> >     - In **Fig.3** (c), We added additional experiments for MNIST and CIFAR10 datasets, where we train the model with the **full dataset** ($m=60000$ for MNIST and $m=50000$ for CIFAR10) and evaluate the *expected log density* with $10000$ test data points. We can observe the **same trade-off** on both BPD and the proposed bound.
> >
> > * Miscommunication, visual inspection for overfit/under-fit, not data quality
> >
> >     - Sorry for the misleading word "quality of generated data" used. We have revised the paper and present "we observe a trade-off between noise and duplicate (or entirely black/white) images in the generated data with respect to the diffusion time T across both datasets", which can reflect the varying of generalization.
> >     -  As claimed in [3], "**average log-likelihood**, Parzen window estimates, and **visual ﬁdelity** of samples—are largely independent of each other when the data is high-dimensional." Our results in Fig.3 do not contradict this conclusion. In fact, the visual fidelity in Fig.3 (a) increases w.r.t the growth of diffusion time $T$ with less noise, but we observe more duplicate images or pure black or white images, indicating more prone to overfitting.
> >
> > ---
> > > **Q1. Intuition for upper bound in theorem 6.3**
> >
> >  * Since the score function $s_{\theta}$ is learned from the train data set. Hence, each single train data point will contribute to the generated sample, reflected on the sample-wise conditional mutual information (CMI) term.
> >  * Theorem 6.3 first upper bound the sample-wised CMI by the conditional mutual information between the generated sample and the whole train data set using information chain rules.
> >  * Then, since the discrete backward process is a Langevin dynamics, we use the technique in [2] to obtain the refined upper bound tailored to the sampling process.
> >  * Intuitively, we can consider the score function contains some information in the train data; the information of the train data is added to the sampled output at each backward step. With the accumulation of information through a large number of steps, the output could overfit the train data.
> >
> > ---
> > > **Q2. Sample size and $T_3$.**
> >
> >  * According to $T_3 = \frac{R \sqrt{(\beta_1 - \beta_0)L^2T^2 + ((1+\tau)\beta_0 - \tau\beta_1)L^2 T)}}{\sqrt{m}}$, we have a common sample complexity $O(\frac{1}{\sqrt{m}})$. This term vanishes only when $m→∞$. We say it will remain non-zero in the context of saying that when $T → ∞$, $T_1$, and $T_2$ converge to zero, while $T_3$ will not. It will increase linearly with the growth of $T$.
> > * Use of enough samples
> >      * Yes, given fixed $T$, a score model trained on large data set (large $m$) with enough samples will have smaller $T_3$. This is similar to supervised learning, where a model trained on a large dataset is less prone to overfitting.
> >      * However, this is not related to the estimation quality of KL-divergences in the upper bound (right-hand side of Theorem 6.2), which can be computed explicitly because the distributions are Gaussians or a mixture of Gaussians. Hence, the estimation of these KL-divergences will not affect the contribution of $T_3$.
> >      *  The sample size largely affects the estimation of  KL divergence $D_{KL}(P_X\|\|Q_{G}^{\pi})$ on the left hand, which requires large amount of test data for accurate estimation in high dimensional setting.
> > ---
> > > **Q3.** Final claim of the theorem in the last lines of the proof.
> >
> > * Thanks for the suggestion. We have added a step-by-step derivation in Appendix C.1 in the revised submission.
> >
> >
> > ---
> > ### References
> > * [1] Zhang and Chen 2023 "Fast Sampling of Diffusion Models with Exponential Integrator.
> >
> > * [2] Pensia, Ankit, Varun Jog, and Po-Ling Loh. "Generalization error bounds for noisy, iterative algorithms." 2018 IEEE International Symposium on Information Theory (ISIT). IEEE, 2018.
> >
> > * [3] Theis et al. 2015 "A note on the evaluation of generative models".

---

> > > ### Comment · Reviewer_o393 · 2024-11-25
> > >
> > > Thank you for your clarifications and for adding results using a fixed step-size.
> > > I have increased my score to a 6.

---

> > > > ### Author Response · Authors · 2024-11-25
> > > >
> > > > Dear Reviewer o393,
> > > >
> > > > Thank you for your insightful suggestions, which have greatly contributed to improving the quality of our paper. We also appreciate the increased score.

---

### Official Review · Reviewer_huQG · 2024-10-24

**Soundness:** 3
**Presentation:** 2
**Contribution:** 2
**Rating:** 6
**Confidence:** 4

**Summary:**

In this paper, the authors explore the theoretical generalization behaviour for both variational autoencoders (VAEs) and diffusion models (DMs), noting that a DM can be seen as an infinite-layered hierarchical VAE. For both models, the behaviour of both the encoder and decoder in the VAE, and forward and reverse diffusion steps in the DM are analyzed. The authors then provide a theoretical upper bound for DMs with respect to the number of diffusion steps and then empirically verify it on several datasets, both synthetic and real (MNIST and CIFAR10), demonstrating its validity and usefulness in training optimal DMs.

**Strengths:**

This paper provides a very detailed and comprehensive information-theoretic analysis of generalization in VAEs and DMs along with experiments that empirically validate it. In particular, the incorporation of encoder-decoder / forward-reverse process into the analysis provides a novel view into their impact on the generative models' generalization behaviour, such as the finding that longer diffusion steps do not necessarily result in better estimates in DMs.

**Weaknesses:**

The paper's writing made it difficult to process the main contributions to the paper for two main reasons:

(1) Despite the abstract suggesting that the VAE's generalization behaviour is studied, much of the paper's focus is on analyzing DM behaviour.

(2) There is notably no experiments that validate VAE behaviour, which suggests that the VAE is studied here as a precursor to understanding the generalization behaviour of DMs.

**Questions:**

I suggest the authors make the writing more clear by including a separate background section on the relationship between VAEs and diffusion models; this will make it clear that the main contribution is providing a theoretical upper-bound for diffusion models as aided by the analysis of VAE generalization. A good place to start is the Variational Diffusion Models paper [1] that explicitly makes this connection by formulating the diffusion learning algorithm in terms of a variational lower bound.

References
[1] Diederik P. Kingma, Tim Salimans, Ben Poole, & Jonathan Ho. (2023). Variational Diffusion Models.

---

> ### Author Response · Authors · 2024-11-24
> **Part 1- Weakness 1 & 2**
>
> We sincerely thank the reviewer for the valuable and constructive feedback. Below, we address your concerns in detail.
>
> ---
> > **W1. Insufficient analysis of VAE's generalization behavior.**
>
> Thanks for the suggestion! We have dedicated more attention to the diffusion model, as their results are relatively underexplored. Consequently, we only highlighted our *theoretical improvements to the previous works for VAEs* in **Theorem 5.1**. However, we agree that additional analysis of VAEs could enhance the paper's completeness and impact. To address this, **we have revised the paper accordingly. In Section D.1 of the Appendix**, we provide a detailed discussion of the prior PAC-Bayes bounds to strengthen the analysis.
>
> - **Explicit Analysis**
>     * As discussed in Sec.6.5.2 [1], the mutual information bound is tighter than the PAC-Bayes bound in expectation. For the more concise form, we transform the "Catoni-style" PAC Bayes bound (Theorem 5.2) in [2] to the expectation bound by integrating over the high-probability guarantee, which gives the following bound that holds for **any** $E_{\phi}(x)$ and some **fixed** $G_{\theta}(z)$:
> $$
> D\_{W\_1}(P\_X\|\|Q\_{G_{\theta}}^\pi) \leq \mathbb{E}\_S L\_{\hat{P}\_X}(E\_{\phi}, G\_{\theta}) + (\frac{\Delta}{\sqrt{2}} + K_{\theta})\sqrt{\frac{1}{m}\sum_{i=1}^m \mathbb{E}\_{X_i} D\_{KL}(E\_{\phi}(X\_i)\|\|\pi)}
> $$where $\Delta:=sup_{x,x^\prime}d(x,x^\prime)$ is the diameter of the bounded input space, $K_{\theta}$ is the Lipschitz constant of $\mu_{\theta}$.
>     * The bounded support assumption w.r.t $\|\|\cdot\|\|$ implies sub-Gaussian with $R = \frac{\Delta}{2}$. Hence, our bound equals the following, which holds for **any** $E_{\phi}(x)$ and **any** $G_{\theta}(z)$:
>   $$D\_{W\_1}(P\_X\|\|Q\_{G\_\theta}^\pi) \leq \mathbb{E}_S L\_{\hat{P}\_X}(E\_{\phi}, G\_{\theta}) + \frac{\Delta}{\sqrt{2}}\sqrt{\frac{1}{m} \sum\_{i=1}^m \mathbb{E}\_{X_i}[D\_{KL}(E\_{\phi}(X_i)\|\|\pi})] + \frac{\Delta}{\sqrt{2}} \sqrt{\frac{1}{m} \sum\_{i=1}^m I(\hat{X}\_i; X\_i| Z_i)}\,.$$
>     * Ignoring the additional generalization term for $G_{\theta}$, our bound is **tighter** than previous work, which has a **smaller factor (without $K_{\theta}$)** over the generalization term for the encoder. Moreover, not all sub-Gaussian random variables are bounded, so our assumption is more **flexible** and valid for unbounded support.
> ---
>  > **W2. Lack of experiments to verify VAE's generalization behavior**
>
>  Thanks for the insightful suggestion. Due to the space limit, we have included the experimental results in the Appendix and highlighted key findings in the discussion section of the main paper.
>
> * **Experiments for VAE**
>     * We added experiments for VAE on the MNIST dataset. The estimated bounds and test loss are plotted in Fig. 4 (a). The previous PAC Bayes bound estimated with train data could fall below the test loss because it does not hold for any $G_{\theta}$. In contrast, our bound estimated on train data well aligns with the test loss because it holds for any $G_{\theta}$ by considering its generalization term.
>
>     * To further illustrate the effectiveness of using the conditional mutual information term to capture the generalization of the generator. We compare its estimation to the **memorization score** proposed in [3], which measures how much more likely an observation is when it is included in the training set compared to when it is not. The result is presented in Fig. 4(b), where we plot the $95\%$ quantile of the memorization score and the conditional mutual information term with respect to the training epochs. The two terms are highly correlated with similar evolving trends, suggesting our bond may capture the memorization to some extent. To be noted, the evaluation of the memorization score requires an *additional validation set*, while our bound only uses the train set.

---

> ### Author Response · Authors · 2024-11-24
> **Part 2 -- Question 1**
>
> > **Q1.  Background section for relationship between VAEs and Diffusion Models.**
>
> Thanks for the suggestion.
> * First off, we would like to clarify some possible misunderstandings. The theoretical results of diffusion models in this paper are **not based on the analysis of VAE**.
> * We proposed a **generalized** analysis framework for *encoder-generator structure models*, where the encoder $E$ and generator $G$ can be any randomized mapping. VAEs and diffusion models are two specific instances of this kind of model. VAE use specific distribution families for $E$ and $G$, e.g., Gaussians. Meanwhile, diffusion models use SDEs or Langevin dynamics as $E$ and $G$.
>
> * In **Figure 1**, we have illustrated the difference in treating Diffusion Models as (1) Hierarchical VAE ([4, 5, 6]) via variational lower bound and (2) time-dependent mappings.
>
> * We have also discussed in **Appendix B** that the general optimization objective $\inf_{E} D\_{KL}(P_X E(X)\|\|G(Z)Q_Z)$ has the following two decompositions, which correspond to (1) VAE (or Hierarchical VAE) and (2) time-dependent mapping (score-based diffusion model), respectively.
>
>      - (1) $\inf\_{E} \left[\mathbb{E}\_{X\sim P\_X} \left(\mathbb{E}\_{Z\sim E(X)} \left[-\log q\_G(X|Z)\right] + D\_{KL}(E(X)\|\|Q_Z)\right)- h(P_X)\right]$
>      - (2) $\inf_{E} D\_{KL}(P^E_Z\|\|Q_Z) + \mathbb{E}\_{Z\sim P^E_Z} D\_{KL}(P_E(X|Z)\|\|G(Z))$,
>        where for $E_T$ and $G_T$ the second term is upper bounded by the path measure and can be further upper bounded by the score matching objective via Girsanove theorem, as discussed in [7].
>
> * **In this paper, we study the diffusion model with (2) instead of (1).**
>
> * However, adding the discussion on the relationship between diffusion models and VAEs via objective (1) is also interesting. **We have included the derivation by formulating diffusion models as hierarchical VAEs with variational decomposition in Appendix B.2 to complete the background.**
>
> ---
> ### References
> - [1] Alquier, Pierre. "User-friendly introduction to PAC-Bayes bounds."_Foundations and Trends® in Machine Learning_ 17.2 (2024): 174-303.
>
> - [2] Sokhna Diarra Mbacke, Florence Clerc, and Pascal Germain. Statistical guarantees for variational autoencoders using pac-bayesian theory. Advances in Neural Information Processing Systems, 36, 2024.
>
> - [3] Van den Burg, Gerrit, and Chris Williams. "On memorization in probabilistic deep generative models." Advances in Neural Information Processing Systems 34 (2021): 27916-27928.
>
> - [4] Diederik P. Kingma, Tim Salimans, Ben Poole, & Jonathan Ho. (2023). Variational Diffusion Models.
> - [5] Chin-Wei Huang, Jae Hyun Lim, and Aaron C Courville. A variational perspective on diffusion-based
> generative models and score matching. Advances in Neural Information Processing Systems, 34: 22863–22876, 2021.
> - [6] Belinda Tzen and Maxim Raginsky. Neural stochastic differential equations: Deep latent gaussian models in the diffusion limit. arXiv preprint arXiv:1905.09883, 2019.
> - [7] Yang Song, Jascha Sohl-Dickstein, Diederik P Kingma, Abhishek Kumar, Stefano Ermon, and Ben Poole. Score-based generative modeling through stochastic differential equations. arXiv preprint
> arXiv:2011.13456, 2020b.

---

> ### Author Response · Authors · 2024-11-26
>
> Dear Reviewer huQG,
>
> Thank you for your efforts in reviewing our paper. As we approach the final day for uploading a revised PDF, we would like to address a few points.
>
> We hope our efforts to include a detailed theoretical comparison and additional experiments for VAE in the rebuttal have successfully addressed your concerns. Furthermore, we added a background section to provide more context and make it easier for reviewers to understand our contributions.
>
> If you have any additional comments or suggestions that may require further modifications to the PDF, please let us know. We would be happy to incorporate your feedback promptly.
>
> Best,
>
> The authors

---

> > ### Author Response · Authors · 2024-11-30
> > **Thanks**
> >
> > Dear Reviewer huQG,
> >
> > Thank you for your time and effort in reviewing our work. With only <= 3 days remaining, we kindly request your feedback on our rebuttal. If any part of our explanation is unclear, please let us know. We would greatly appreciate it if you could confirm whether your concerns have been addressed. If the issues are resolved, we would be grateful if you could consider reevaluating the work. Should you need any further clarification, we are happy to provide it promptly before the discussion deadline.
> >
> > Best,
> >
> > The authors

---

> > > ### Comment · Reviewer_huQG · 2024-12-02
> > >
> > > First of all, I apologize for the delay.
> > >
> > > While I am not entirely convinced by the direction of the paper, the additional explanation makes the paper more clear to understand. As such, I will be increasing my score accordingly. Thank you for the clarifications.

---

> > > > ### Author Response · Authors · 2024-12-02
> > > > **Thanks**
> > > >
> > > > Dear Reviewer huQG,
> > > >
> > > > Thank you for taking the time to review our rebuttal and for updating your score. We sincerely appreciate your thoughtful reconsideration and the effort you put into evaluating our work.
> > > >
> > > > Best,
> > > >
> > > > The Authors

---

### Official Review · Reviewer_qVWA · 2024-10-27

**Soundness:** 3
**Presentation:** 4
**Contribution:** 3
**Rating:** 8
**Confidence:** 4

**Summary:**

This paper derives generalization bounds for encoder-generator-based generative models, considering both the encoder and generator components. The analysis is specifically applied to Variational Autoencoders (VAEs) and Diffusion Models (DMs), with tailored generalization bounds presented for each. Notably, a novel trade-off relationship is introduced for DMs. The theoretical results are supported by numerical experiments, demonstrating that the derived bounds offer practical utility, particularly in capturing the trade-offs in generalization.

**Strengths:**

- This paper derives a generalization bound for encoder-generator architectures under the relatively mild assumption of sub-Gaussian loss functions. As noted in lines 286-291, the paper provides an intuitive explanation of these bounds and a convincing discussion of the trade-offs involved.
- Corollaries 4.2 and 4.3 extend the analysis to evaluate the Wasserstein distance and KL divergence between the generative model's distribution and the data distribution, offering valuable tools for the theoretical analysis of a wide range of generative models.
- For VAEs, the paper presents a bound that applies to arbitrary generators, distinguishing it from existing research.
- In the case of DMs, a new trade-off relationship is derived. Under certain reasonable assumptions, the generalization bound involving mutual information of $G^{θ}$ is established concerning sample size, score function bounds, and time $T$. This leads to a tighter $ m^{-1/2}$ bound.
- The results are further substantiated by numerical experiments that validate the theoretical trade-offs observed in the upper bound.

**Weaknesses:**

While the results are significant in terms of learning theory by considering the effects of both the encoder and generator, some areas could be further improved:
- Although challenging, the analysis does not incorporate the complexity of the learning models. Including bounds related to the complexity of simple neural networks or linear models could strengthen the work.
- Aside from the theoretical analysis provided by the generalization bound, it would be beneficial to relate these results to those from sharp theoretical analyses in proportional limit settings. For example, in denoising encoders [1], autoencoders [2], or variational autoencoders [3, 4], the generalization error has been sharply evaluated by fixing the ratio $\alpha = n/d = \Theta(1)$, where $n \to \infty$ is the number of data points and $d \to \infty$ is the data dimension. Including such discussions in the related work section would provide a more comprehensive view of the literature.
- It would be interesting to see the results of training on the complete datasets of MNIST and CIFAR-10 rather than in a few-shot setting.
- The paper refers to the claime by Thesis et al. (2015) that "accurately estimating the KL divergence and Bits-Per-Dimension (BPD) for high-dimensional data distributions with limited data is challenging". However, this could be addressed using population MCMC or other Markov chain Monte Carlo methods. Investigating whether the trade-offs as in the upper bound remain evident in such scenarios would be intriguing.

- [1]: Cui, Hugo, and Lenka Zdeborová. "High-dimensional asymptotics of denoising autoencoders." Advances in Neural Information Processing Systems 36 (2023): 11850-11890.
- [2]: Refinetti, Maria, and Sebastian Goldt. "The dynamics of representation learning in shallow, non-linear autoencoders." International Conference on Machine Learning. PMLR, 2022.
- [3]: Ichikawa, Yuma, and Koji Hukushima. "Dataset size dependence of rate-distortion curve and threshold of posterior collapse in linear vae." arXiv preprint arXiv:2309.07663 (2023).
- [4]: Ichikawa, Yuma, and Koji Hukushima. "Learning Dynamics in Linear VAE: Posterior Collapse Threshold, Superfluous Latent Space Pitfalls, and Speedup with KL Annealing." International Conference on Artificial Intelligence and Statistics. PMLR, 2024.

**Questions:**

- The paper recommends adding terms for the generalization of $G$ to provide insights and practical guidance for VAEs. However, estimating these terms is generally considered difficult. Are there any methodologies that could facilitate this estimation?
- The few-shot setting with $m=16$ is mentioned in the experiments on real data. Could you provide more details on this experimental setup? Does it involve generating new data with a pre-trained model in a few-shot learning manner, or is it simply trained with limited data?
- What is the definition of $g$ in line 209?
- How would the generalization bounds change if the data were assumed to lie on a low dimensional manifold? What predictions can be made regarding such cases?

---

> ### Author Response · Authors · 2024-11-24
> **Part 1 --- Weakness 1-3**
>
> We sincerely thank the reviewer for the positive feedback, valuable comments and constructive suggestions. Below are our responses.
>
> ---
> > **W1. Compare to sharp theoretical analyses in proportional limit settings.**
>
>   Thanks for suggesting these missing related works, which are relevant. We have inclucded the following discussion in the related work section, please check the updated submission.
>
>   * >Another approach involves deriving exact formulae under specific data distributions and high-dimensional limits. Assuming $n = \infty$, [2] examines the test error for nonlinear two-layer autoencoders as $d \to \infty$. [1] investigates the generalization error for nonlinear two-layer denoising auto-encoders when $\alpha = \frac{n}{d} = \Theta(1)$. Focusing on the spiked covariance model and linear $\beta$-VAEs, [3] analyzes generalization error with SGD dynamics, where fixed-point analysis reveals posterior collapse when $\beta$ exceeds some threshold, suggesting appropriate KL annealing to accelerate convergence. [4] uses the Replica method to derive asymptotic generalization error for $\alpha = \frac{n}{d} = \Theta(1)$, showing a peak in error at small $\beta$, which disappears after $\beta$ beyond some threshold. This can lead to posterior collapse regardless of the dataset size.'''
>
> ---
> > **W2.  Include complexity of simple neural networks or linear models.**
>
> * We consider simple linear VAE models following the suggested related work [3, 4] and let $E_{\phi}(x)=N(\mu_{\phi}(x), \sigma^2 I_{d^\prime}))$ with $\mu_{\phi}(x) = \frac{1}{\sqrt{d}}\phi^Tx, \phi=R^{d \times d^\prime}$ and $G_{\theta}(z) = N(\mu_{\theta}(z), I_d)$ with $\mu_{\theta}=\frac{1}{\sqrt{d}}\theta^T z, \theta \in R^{d^\prime\times d}$,  $\pi=N(0, I_{d^\prime}), x\in R^d, z \in R^{d^\prime}$.  Combining the estimation for the conditional mutual information term, we proposed in Appendix D.2, where a duplicate randomly initialized generator $G_{\tilde{\theta}}(z) = N(\mu_{\tilde{\theta}}(z), I_d)$ is needed. Then, we have the following generalization term:
> $$
> \sqrt{2} R \sqrt{\frac{1}{2}(\sigma^2 - 1)d^\prime - d^\prime \log(\sigma^2) + \frac{1}{d} \mathbb{E}\_S\left(\frac{1}{m} \sum\_{i=1}^m x_i^\top \phi \phi^\top x_i\right)+ \frac{\sigma^2}{2dm} \|\| \theta - \tilde{\theta} \|\|^2}.
> $$
>
> * Consider the propotional limit setting with $\alpha=\frac{m}{d}=\Theta(1)$, we have the following two major terms: $\sqrt{\frac{\alpha}{m}\mathbb{E}\_S\left(\frac{1}{m} \sum\_{i=1}^m x_i^\top \phi \phi^\top x_i\right)}$ for encoder and $\sqrt{\frac{\sigma^2\alpha}{2m^2} \| \|\theta - \tilde{\theta} \|\|^2}$ for generator.
>
> * More detailed analysis w.r.t the convergence given specific optimization algorithm is left as future work.
>
>  ---
>
> > **W3.  Results of training on the complete datasets of MNIST and CIFAR-10 rather than in a few-shot setting.**
>
> Thanks for the suggestion!
>
>  - In **Fig.3** (c), we added an additional experiments for MNIST and CIFAR10 datasets, where we train the model with the **full dataset ($m=60000$ for MNIST and $m=50000$ for CIFAR10)** and evaluate the *expected log density* with **$10000$ test data** points. We can observe the **same trade-off** on both BPD and the proposed bound.
> ---

---

> > ### Author Response · Authors · 2024-11-24
> > **Part 2 -- Weakness 4 & Question 1**
> >
> > > **W4. Estimate the KL divergence/log density with MCMC methods.**
> >
> >  Thanks for the suggestion! The estimation of log density is widely considered a hard problem [6, 7, 8].
> >  * In this paper, we have considered **estimating the expected log density** with $n$ test data sampled from $P_X$ and the approximated density function of generator $G$, where we use the approximated density function proposed by [9]. The sample complexity in this case is $O(\frac{V}{\sqrt{n}})$, where $V$ is the variance of $\log q_G(x)$ under $P_X$.
> >
> >  * In the revised submission, we can observe the trade-off for BPD exits when estimated using **$10000$ test samples** for models **trained on $m=50000$ data points.** In this case, $G_{\theta}^\pi$ and $P_X$ are similar and have large overlap, $V$ is small. So, the estimation of BPD is accurate in capturing the trade-off.
> >
> >  * However, in the original submission,  $G$ is trained on few-shot data ($m=16$), where the supports of $P_X$ and $Q_G^{\pi}$ have poor overlap and the variance $V$ will be large, indicating more test samples are needed for an accurate estimation. In **Fig.~ 7(b)**, we use $10000$ test samples to estimate the log density for the few-shot training setting ($m=16$), where BPD still cannot capture the trade-off, and we observe large variance $V$ on the plot.
> >
> >  * MCMC methods such as Annealed Importance Sampling (AIS) and Bidirectional Monte Carlo [6] can help reduce variance by introducing a sequence of intermediate distributions. However, their effectiveness depends heavily on the careful design of these intermediate distributions and the associated transition kernels, especially in high-dimensional settings. Previous studies have applied these methods to estimate log densities for fast-sampling models like GANs and VAEs trained on full datasets [6,8]. However, it remains unclear how well these techniques will perform in extreme few-shot scenarios ($m=16$) or how to efficiently adapt them for diffusion models, which typically have slower sampling speed. Given the limited rebuttal period, we are still working on this and will provide updates if significant progress is made.
> > ---
> > > **Q1. How to estimate the generalization term of G efficiently?**
> >
> >   Thanks for the insightful question.
> >   * By upper bounding the sample-wise conditional mutual information with the conditional mutual information between the generated sample and the train data set, we have proved the following upper bound of the conditional mutual information term in Sec.D.2 in the Appendix:
> >   $$\frac{1}{m}\sum\_{i=1}^m I(\hat{X}\_i; X\_i| Z\_i) \leq \frac{1}{m} \mathbb{E}\_S \left(\sum\_{i=1}^m  \frac{1}{m} \mathbb{E}\_{Z_i\sim E(X_i)}D\_{KL}(G_{\theta}(Z_i)\|\| G_{\tilde{\theta}}(Z_i))\right)\,
> >   $$
> >   where $G_{\tilde{\theta}}$ is the data-free generator distribution. We can simply use another randomly initialized generator with parameter $\tilde{\theta}$. According to specific distribution family of the generator used in VAEs, we can compute the KL-divergence efficiently.
> > * We also provided an additional experiment for VAEs by estimating this bound on the MNIST dataset. The estimated bounds and test loss are plotted in **Fig. 4 (a)**. The previous PAC Bayes bound estimated with train data falls below the test loss because the bound **does not hold for any $G_{\theta}$**. In contrast, our bound estimated on train data well aligns with the test loss because it holds for any  $G_{\theta}$ by introducing the generalization term for the generator.
> >
> >  *  We further compared the estimation of the mutual information term to the memorization score proposed in [5], which measures how much more likely an observation is when it is included in the training set compared to when it is not. The result is presented in **Fig. 4(b)**, where we plot the $95\%$ quantile of the **memorization score** and the conditional mutual information term with respect to the training epochs. The two terms are highly correlated with similar evolving trends, suggesting our bond may capture the memorization to some extent. To be noted, the evaluation of the memorization score requires additional validation sets, while our bound only uses the train set.

---

> > ### Comment · Reviewer_qVWA · 2024-11-25
> > **response**
> >
> > Thank you for your response.
> > I was able to understand all the questions well.
> >
> > I also feel that the related work has made the overall picture of learning theory for generative models easier to grasp.
> >
> > Furthermore, the analysis results for the simple linear VAE are very intriguing!
> > I believe it would be a great addition to include this analysis in the appendix or elsewhere in the camera-ready version.

---

> > > ### Author Response · Authors · 2024-11-25
> > >
> > > Dear reviewer qVWA,
> > >
> > > Thanks for your prompt response and suggestions, especially regarding the related works and the discussion of linear VAEs. We do believe that applying the proposed theory to a tractable linear model is insightful and can help people understand the problems better. Hence, we have added this discussion to Sec. D.3 in the revised submission.

---

> ### Author Response · Authors · 2024-11-24
> **Part 3 --- Question 2&3&4**
>
> > **Q2. Details for the few-shot setting.**
>
> ---
>   * Thanks for asking. We train the model with few-shot data $m=16$ to make sure we can present visual difference within limited random draws, as mentioned in Fig. 6 and Fig. 7.  To estimate the test data kl divergence and the bpd, we use the trained model to generate 100 images and randomly sample 100 test images,  as we discussed in the **Train-Test Details** in Sec. G.2.3 and Sec. G.2.1  in the Appendix.
> ---
> > **Q3. What is the definition of g in line 209?**
>
>   * Sorry for the typo, it's $\lambda$ in stead of $g$.
>
> ---
> > **Q4. Generalization under low dimensional manifold**
>
>   * If the data distribution $P_X$ is assumed to lie on a low dimensional manifold, and if this does not break the sub-gaussian assumption, such assumption will lead to a smaller sub-gaussian parameter.
>
>  * For diffusion model, manifold assumption indicates $p_0(x)≈ 0$ for most $x$, which will lead to a large Lipschitz constant $\|\|∇\hat{p}_t(x)\|\|$ at $t=0$. Hence, we need to add a penalty term to control the Lipschitz term of the score model or stop sampling at $t=\epsilon, \epsilon \in (0, T]$ to avoid an exploding score.
>
>  * For VAEs, we can use this information to select the distribution families for the encoder, generator and the prior $\pi$, which can get tighter KL divergence and improve the performance.
> ---
> ## References
>   - [1]: Cui, Hugo, and Lenka Zdeborová. "High-dimensional asymptotics of denoising autoencoders." Advances in Neural Information Processing Systems 36 (2023): 11850-11890.
>
>   - [2]: Refinetti, Maria, and Sebastian Goldt. "The dynamics of representation learning in shallow, non-linear autoencoders." International Conference on Machine Learning. PMLR, 2022.
>
>   - [3]: Ichikawa, Yuma, and Koji Hukushima. "Learning Dynamics in Linear VAE: Posterior Collapse Threshold, Superfluous Latent Space Pitfalls, and Speedup with KL Annealing." International Conference on Artificial Intelligence and Statistics. PMLR, 2024.
>
>   - [4]: Ichikawa, Yuma, and Koji Hukushima. "Dataset size dependence of rate-distortion curve and threshold of posterior collapse in linear vae." arXiv preprint arXiv:2309.07663 (2023).
>
>
>   - [5] Van den Burg, Gerrit, and Chris Williams. "On memorization in probabilistic deep generative models." Advances in Neural Information Processing Systems 34 (2021): 27916-27928.
>
>   - [6] Wu, Y., Burda, Y., Salakhutdinov, R., & Grosse, R. (2016). On the quantitative analysis of decoder-based generative models. arXiv preprint arXiv:1611.04273.
>
>   - [7] Theis et al. 2015 "A note on the evaluation of generative models".
>
>   - [8] Huang, Sicong, et al. "Evaluating lossy compression rates of deep generative models." International Conference on Machine Learning. PMLR, 2020.
>
>   - [9] Huang, Chin-Wei, Jae Hyun Lim, and Aaron C. Courville. "A variational perspective on diffusion-based generative models and score matching." Advances in Neural Information Processing Systems 34 (2021): 22863-22876.
>   - [10] Chatterjee, Sourav, and Persi Diaconis. "The sample size required in importance sampling." The Annals of Applied Probability 28.2 (2018): 1099-1135.

---

### Official Review · Reviewer_7s19 · 2024-11-04

**Soundness:** 3
**Presentation:** 3
**Contribution:** 2
**Rating:** 5
**Confidence:** 4

**Summary:**

This paper provides a information-theoretic framework of generalization theory for variational auto-encoders and diffusion models. Authors consider generalization properities for both the encoder and the generator in VAEs. Their generalization bounds can be estimated using only training data, providing a practical guidance for hyperparameter selection.

**Strengths:**

- The authors address the critical topic of generalization in generative models, and provide estimable bounds for both the encoder and the generator in VAEs.
- Bounds for VAEs avoid Wasserstein distance and impose milder assumptions (bounded to sub-Gaussian).
- Bounds for DMs overcome the challenges associated with KL-divergence's non-satisfaction of the triangle inequality, and contribute to a clearer understanding of diffusion time’s role in generalization and model performance.

**Weaknesses:**

- In line 98, the paper asserts that the bounds for the encoder are tighter, yet this claim lacks sufficient detail. Although some comparisons to previous bounds are made in line 324, there remains a need for a more explicit, quantitative analysis to illustrate the improvements over existing bounds. Adding a direct comparison or detailed quantitative analysis would make the claim more substantiated and provide clearer evidence of the improvement.
- The proposed generalization bounds do not clearly indicate a dependency on the number of samples, $m$, limiting their practical applicability. To make the bounds more actionable, it would be helpful if the authors explicitly stated the sample complexity (e.g., $O(1/ m)$) within the main theorems or discussed the bounds’ scaling with respect to $m$. This addition could guide readers in understanding the bounds’ robustness and sample efficiency.
- The paper suggests a theoretical trade-off for diffusion models based on diffusion time, but this trade-off is not consistently reflected in the experimental results, as KL and BPD metrics do not show this effect. The authors could address this discrepancy by either providing a potential explanation for the divergence between the theoretical and experimental findings or suggesting additional experiments that might better capture the trade-off. Revisiting this section would enhance clarity and ensure its alignment with the paper’s core contributions.

**Questions:**

- The generalization bounds resemble Theorem 4.1, which primarily links encoder mappings' complexity to the gap between empirical generalization and expected generalization. But this should also affects empirical generalization. Can you provide a theorem telling how this two terms together affects expected generalization?
- As far as I know, in practical works of generative models, the target distribution is always unknown, which means the empirical generalization error is always unknown. Could the authors elaborate on the practical benefits of bounding the gap between expected and empirical generalization error, particularly when the empirical error itself may not be measurable?
- As mentioned in line 350, generalization bounds are computable, can you explain how to compute term $\hat{L}_{ESM}$ among upper bounds in Theorem 6.2?

---

> ### Author Response · Authors · 2024-11-24
> **Part1 -- Weakness 1 & 2**
>
> We sincerely thank the reviewer for the valuable and constructive feedback. Below, we address your concerns in detail.
>
> ---
> >**W1. More explicit, quantitative analysis to illustrate the improvements over existing bounds.**
>
> Thanks for the suggestion, we have modified the paper accordingly. In Sec. D.1 in the Appendix, we provide a detailed discussion w.r.t previous PAC Bayes bound.
> 1. **Explicit Analysis**
>     * As discussed in Sec.6.5.2 [1], the mutual information bound is tighter than the PAC-Bayes bound in expectation. For the more concise form, we transform the "Catoni-style" PAC Bayes bound (Theorem 5.2) in [2] to the expectation bound by integrating over the high-probability guarantee, which gives the following bound that holds for **any** $E_{\phi}(x)$ and some **fixed** $G_{\theta}(z)$:
> $$
> D\_{W\_1}(P\_X\|\|Q\_{G_{\theta}}^\pi) \leq \mathbb{E}\_S L\_{\hat{P}\_X}(E\_{\phi}, G\_{\theta}) + (\frac{\Delta}{\sqrt{2}} + K_{\theta})\sqrt{\frac{1}{m}\sum_{i=1}^m \mathbb{E}\_{X_i} D\_{KL}(E\_{\phi}(X\_i)\|\|\pi)}
> $$where $\Delta:=sup_{x,x^\prime}d(x,x^\prime)$ is the diameter of the bounded input space, $K_{\theta}$ is the Lipschitz constant of $\mu_{\theta}$.
>     * The bounded support assumption w.r.t $\|\|\cdot\|\|$ implies sub-Gaussian with $R = \frac{\Delta}{2}$. Hence, our bound equals the following, which holds for **any** $E_{\phi}(x)$ and **any** $G_{\theta}(z)$:
>   $$D\_{W\_1}(P\_X\|\|Q\_{G\_\theta}^\pi) \leq \mathbb{E}_S L\_{\hat{P}\_X}(E\_{\phi}, G\_{\theta}) + \frac{\Delta}{\sqrt{2}}\sqrt{\frac{1}{m} \sum\_{i=1}^m \mathbb{E}\_{X_i}[D\_{KL}(E\_{\phi}(X_i)\|\|\pi})] + \frac{\Delta}{\sqrt{2}} \sqrt{\frac{1}{m} \sum\_{i=1}^m I(\hat{X}\_i; X\_i| Z_i)}\,.$$
>     * Ignoring the additional generalization term for $G_{\theta}$, our bound is **tighter** than previous work, which has a **smaller factor (without $K_{\theta}$)** over the generalization term for the encoder. Moreover, not all sub-Gaussian random variables are bounded, so our assumption is more **flexible** and valid for unbounded support.
>
> 2. **Quantitative Analysis**
>     * We add experiments for VAE on the MNIST dataset based on [3]. The estimated bounds and test loss are plotted in **Fig. 4 (a)**. The previous PAC Bayes bound estimated with train data falls below the test loss because the bound **does not hold for any $G_{\theta}$**. In contrast, our bound estimated on train data well aligns with the test loss because it holds for any $G_{\theta}$ by considering the generalization term for $G_{\theta}$.
>     * To further illustrate the effectiveness of using the conditional mutual information term to capture the generalization of $G_{\theta}$. We compare its estimation to the **memorization score** proposed in [3], which measures how much more likely an observation is when it is included in the training set than when it is not. The result is presented in **Fig. 4(b)**, where we plot the $95\%$ quantile of the memorization score and the conditional mutual information term along the training epochs. The two terms are highly correlated with similar evolving trends, suggesting our bond may capture the memorization to some extent. To be noted, evaluating the memorization score requires an additional validation set, while our bound only uses the train set.
> ---
> > **W2. Dependency on the number of samples $m$.**
>
> * Thanks for suggestion. The general bound in Theorem 4.1 is algorithm-dependent and data-dependent, which could be used to analyze encode-generator structure algorithms, not limited to VAEs and diffusion models. Thus, the sample complexity depends on the specific algorithm and data distribution. However, the sample-wised mutual information term is often sublinear to $m$ even though it appears not to depend on $m$ directly.
>
>   - For VAEs, we have proved the following upper bound of the conditional mutual information term in Sec.D.2 in the Appendix:
>   $$\frac{1}{m}\sum\_{i=1}^m I(\hat{X}\_i; X\_i| Z\_i) \leq \frac{1}{m} \mathbb{E}\_S \left(\sum_{i=1}^m  \frac{1}{m} \mathbb{E}\_{Z_i\sim E(X_i)}D\_{KL}(G\_{\theta}(Z\_i)\|\| G\_{\tilde{\theta}}(Z\_i))\right) \in \mathcal{O}(\frac{1}{m})\,
>   $$
>   where $G_{\tilde{\theta}}$ is the data-free generator distribution. Hence, the sample complexity of our bound is $O(1/\sqrt{m})$.
>   - For Diffusion Models, we have discussed the sample complexity is $O(1/\sqrt{m})$ in Theorem 6.3 in the main paper.
> ---

---

> > ### Author Response · Authors · 2024-11-24
> > **Part 2 -- Weakness 3**
> >
> > > **W3. Additional experiments and explanation on the empirical results for the diffusion time trade-off.**
> >
> >   That's a good point.
> >   * Clarification of our experimental settings
> >     - First, we would like to clarify that the trade-off is not consistently reflected by KL and BPD when the data is **high-dimensional** and when using **few-data** to train the diffusion model.
> >     - For the simple 2D Swissroll data,  the trade-off is consistently reflected on all the metrics.
> >     - We previously chose the **few-shot setting with $m=16$**, as it allows for easy visual inspection of potential overfitting or memorization in the generated samples. In **Fig. 3 (b,c).** of the original submission (now plotted on the same figure in Fig.3(b)), we estimate the KL divergence and log density using **100 test data points**.
> >
> >   * Additional experiments better capture the trade-off.
> >     - In **Fig.3** (c), we add additional experiments for MNIST and CIFAR10 datasets, where we train the model with the **full dataset ($m=60000$ for MNIST and $m=50000$ for CIFAR10)** and evaluate the *expected log density* with **$10000$ test data** points. In this case, we can observe the **same trade-off** on both BPD and the proposed bound.
> >
> >   * Difficulties in estimating the KL or log density for high dimensional data under the few-shot setting.
> >
> >     - Recall the definition of KL divergence $D_{KL}(P_X\|\|Q_{G}^{\pi})= \int p(x) \log \frac{p(x)}{q_G(x)}dx$. Ignoring the constant $\int p(x)\log p(x) dx$, we consider the **expected log density** $\int p(x) \log q_G(x) dx$.
> >     - The difficulty lies in the unknown density $p(x)$ and we may not have access to $q_G(x)$ for implicit models. VAEs and DMs are often considered explicit models where $q_G(x)$ can be approximated by using methods like the variational lower bound (VLB) or numerical integration of the reverse process.
> >     - How much data do we need to accurately estimate these terms?
> >       1. **Estimate the KL divergence** through non-parametric estimation with test data sampled from $P_X$ and generated samples from diffusion models.
> >          This process involves sampling $n$ test data points from $P_X$ and $\tilde{n}$ data points from $Q_G^{\pi}$, then estimate their densities $p(x)$ and $q_G(x)$ using methods like KDE, and finally obtain $\hat{D}\_{KL}(P_X \|\|Q_{G}^{\pi})=\frac{1}{n}\sum_{i=1}^n(\log p(x_i) - \log q_G(x_i))$.
> >
> >          For KDE, the sample complexity for estimating the densities is $O(n^{-\frac{4}{d+4}} + {\tilde{n}}^{-\frac{4}{d+4}})$ [4], which suffers the curse of dimensionality when $d$ is large. In our experiments, we have $d=789$ for MNIST and $d=1024$ for CIFAR10. So it requires generating numerous samples with diffusion models to do this kind of estimation, which is extremely time-consuming. Moreover, KDE itself can become computationally intensive for large datasets due to the pairwise distance calculations involved. We hence dropped the KDE estimation in the additional experiments, where we use large test data and only estimate the expected log densities.
> >
> >       2. **Estimate the expected log density** with $n$ test data sampled from $P_X$ and the approximated density function of generator $G$.
> >         The sample complexity in this case is $O(\frac{V}{\sqrt{n}})$.
> >         When $G$ is trained on few-shot data ($m=16$),
> >         the supports of $P_X$ and $Q_G^{\pi}$ will have poor overlap, the variance $V$ of $\log q_G(x)$ under $p(x)$ will be large, indicating more test samples are needed for an accurate estimation.
> >         In **Fig.~ 7(b)**, we use $10000$ test samples to estimate the log density for few-shot training setting, where BPD still cannot capture the trade-off and we observe large variance $V$ on the plot. However, In the revised submission, we can observe the trade-off for BPD exits when estimated using $10000$ test samples because the model is trained on $m=50000$ data points, which gives a small $V$.

---

> ### Author Response · Authors · 2024-11-24
> **Part 3 -- Question 1 & 2 & 3**
>
> >**Q1 empirical generalization and expected generalization**
>
>   * If we understand correctly, the empirical generalization mentioned by the reviewer is the **empirical reconstruction error** and the expected generalization error is the **generation error** in the paper. Their expected gap is the left hand side of the generalization bound in Theorem 4.1.
>   * Yes, both the **encoder mapping** and **generator mapping** affect the generalization gap and the the empirical reconstruction error.
>     - Theorem 4.1 also gives $\mathbb{E}\_S L\_{P_X}^{\pi}(E, G)\leq \mathbb{E}\_S L_{\hat{P}_X}(E, G) + \text{generalization terms}$.
>   * Corollary 4.2 and 4.3 can tell us how these two terms affect the expected generalization. In the proof of Corollary 4.2, we have proved that the Wasserstein distance is equivalent to the generation error (expected generalization) for L2-norm loss. In the proof of Corollary 4.2, we have proved that the KL divergence is upper bounded by the generation error minus the entropy $h(P_X)$.
>   * If the reviewer means to analyze the optimization error of the reconstruction loss and provide excess risk analysis for all the right-hand sides, this would be hard for deep networks where the optimization objective is non-convex. The analysis needs to specify algorithms and additional assumptions, such as the smoothness of the loss function, which is beyond this paper's scope. We left this as future work, but thanks for the suggestion; this is a promising direction for a more refined analysis.
>  ---
>  > **Q2. Unknown empirical generalization**
>
>    * We agree the target distribution is always unknown. However, empirical reconstruction error is known because it is defined based on the sampled train data. Note the empirical distribution is known based on its definition $\hat{P}_X=\frac{1}{m}\sum\_{i=1}^m \delta\_{X_i}$. The expectation w.r.t to $S$ is estimated by running the algorithm for several random draws of $S$. This is a common approach in information-theoretic learning literature.
> ---
> >**Q3. How is $\hat{L}_{ESM}$ estimated?**
>
>   * The empirical score matching loss $\hat{L}\_{ESM}(\theta, \lambda(\cdot))$ can be directly estimated.  Due to computational efficiency, it's usually estimated with the empirical denoising score matching loss $\hat{L}\_{DSM}(\theta, \lambda(\cdot))=\frac{1}{2}\int\_{0}^T \frac{1}{m}\sum\_{i=1}^m \mathbb{E}\_{X_t \sim E_t(X_i)}[\lambda^2(t)\|\|∇\log p\_{E_t}(X_t|X_i) - s\_{\theta}(X_t,t)\|\|_2^2 dt]$, where $p\_{E_t}(X_t|X_i)$ is the gaussian density and the integral over $t$ is approximated with uniform sampling.
>
> ---
> ### References
> - [1] Alquier, Pierre. "User-friendly introduction to PAC-Bayes bounds."_Foundations and Trends® in Machine Learning_ 17.2 (2024): 174-303.
>
> - [2] Sokhna Diarra Mbacke, Florence Clerc, and Pascal Germain. Statistical guarantees for variational autoencoders using pac-bayesian theory. Advances in Neural Information Processing Systems, 36, 2024.
>
> - [3] Van den Burg, Gerrit, and Chris Williams. "On memorization in probabilistic deep generative models." Advances in Neural Information Processing Systems 34 (2021): 27916-27928.
>
> - [4] Tsybakov, Alexandre B., and Alexandre B. Tsybakov. "Nonparametric estimators." Introduction to Nonparametric Estimation (2009): 1-76.
>
> - [5] Wu, Y., Burda, Y., Salakhutdinov, R., & Grosse, R. (2016). On the quantitative analysis of decoder-based generative models. arXiv preprint arXiv:1611.04273.
> - [6] Theis et al. 2015 "A note on the evaluation of generative models".
> - [7] Huang, Sicong, et al. "Evaluating lossy compression rates of deep generative models." International Conference on Machine Learning. PMLR, 2020.

---

> > ### Author Response · Authors · 2024-11-30
> > **Thanks**
> >
> > Dear Reviewer 7s19,
> >
> > Thank you for your time and effort in reviewing our work. With only <= 3 days remaining, we kindly request your feedback on our rebuttal. If any part of our explanation is unclear, please let us know. We would greatly appreciate it if you could confirm whether your concerns have been addressed. If the issues are resolved, we would be grateful if you could consider reevaluating the work. Should you need any further clarification, we are happy to provide it promptly before the discussion deadline.
> >
> > Best,
> >
> > The authors

---

> > > ### Comment · Reviewer_7s19 · 2024-12-03
> > > **Thanks**
> > >
> > > Thank you for the detailed response. My concerns have been partially addressed.

---

> ### Author Response · Authors · 2024-12-03
> **Thanks**
>
> Dear Reviewer 7s19,
>
> Thank you for confirming that we have partially addressed your concerns. We have put significant effort into conducting additional experiments and providing further theoretical interpretations to strengthen our work.
>
> If you feel that most of your concerns have been adequately addressed, we would sincerely appreciate it if you could consider reevaluating the score. That said, we fully understand and respect that the decision to adjust the score rests entirely with you.
>
> If you still have any specific concerns that remain unaddressed, please feel free to let us know. We would be happy to address them during the remaining time available for authors to respond on the forum.
>
> Thank you again for your time and thoughtful feedback.
>
> Best,
>
> The authors

---

### Meta-Review · Area_Chair_3Act · 2024-12-21

**Metareview:**

In this work, the authors leverage information-theoretic tools to obtain novel generalization bounds for encoder and generator of VAEs and diffusion models. Reviews were generally positive, commenting on the relevance of the work, the detailed, clear, and comprehensive exposition, the mild assumptions for the bounds, and how the diffusion model bounds suggest a new trade-off relationship and elucidate a clearer understanding of the role of the diffusion time. Some suggestions for a future revision were provided, but otherwise, all primary concerns were addressed during the discussion period. This is high-quality work that is suitable for publication.

**Additional Comments On Reviewer Discussion:**

Reviewer 7s19 requested additional detail on how the bounds are tighter than prior findings; commented on the implicit dependency on the sample size, reducing practicality; and raised concerns that the trade-off in the bound is not reflected in some of the experiments. The authors addressed each point in turn, and the reviewer was mostly satisfied with the additions. Reviewer qVWA expressed a positive impression of the paper, but commented that the analysis did not incorporate complexities; inquired about the relationship to results obtained in the large data, large dimension limit; asked about training on the entire MNIST, CIFAR10 datasets; and requrested comparisons to density estimation via MCMC. The authors developed a new generalization bound in response, addressing all remaining points. The reviewer raised their score accordingly. Reviewer huQG criticized the limited analysis of the VAE; in response, the authors developed new bounds and performed additional experiments for the VAE to be incorporated into a new revision. The reviewer raised their score in response. Finally, Reviewer o393 expressed concerns about the step size varying, and how Figures 3b,3c do not support the claim (same as Reviewer 7s19). The authors provide a theoretical argument to counter these concerns, and perform experiments with full datasets. This reviewer also increased their score.

---

### Decision · Program_Chairs · 2025-01-22

Accept (Poster)